**Brief Communication**

# MSⁿLib: efficient generation of open multi-stage fragmentation mass spectral libraries

Corinna Brungs [1,2,10], Robin Schmid [1,3,10] ✉, Steffen Heuckeroth [4], Aninda Mazumdar [5], Matúš Drexler[1], Pavel Šácha [1], Pieter C. Dorrestein [3], Daniel Petras[6], Louis-Felix Nothias[7,8], Václav Veverka[1,9], Radim Nencka[1], Zdeněk Kameník [5] & Tomáš Pluskal [1] ✉

Untargeted high-resolution mass spectrometry is a key tool in clinical metabolomics, natural product discovery and exposomics, with compound identification remaining the major bottleneck. Currently, the standard workflow applies spectral library matching against tandem mass spectrometry (MS²) fragmentation data. Multi-stage fragmentation (MSⁿ) yields more profound insights into substructures, enabling validation of fragmentation pathways; however, the community lacks open MSⁿ reference data of diverse natural products and other chemicals. Here we describe MSⁿLib, a machine learning-ready open resource of >2 million spectra in MSⁿ trees of 30,008 unique small molecules, built with a high-throughput data acquisition and processing pipeline in the open-source software mzmine.

Accurate structural elucidation and compound annotation in untargeted mass spectrometry (MS) typically rely on matching fragmentation spectra (MS²⁻ⁿ) against reference libraries[1–3]. However, low annotation rates remain a persistent challenge, largely due to the limited structure coverage of high-quality open spectral libraries compared with the vast known chemical space captured in compound databases such as COCONUT[4], ChEMBL[5] or PubChem[6,7]. Obtaining reference standards, especially purified natural products, is challenging and costly. In multi-stage fragmentation (MSⁿ), precursor ions are selected and fragmented in multiple iterations, producing spectral trees with deeper structural insights[8–10]. While MSⁿ provides deeper structural information than MS², and is crucial for characterizing complex molecules and distinguishing isomers[8,9,11], its widespread application is constrained by the lack of public data. Open MSⁿ (that is, n > 2) spectral entries currently number below 2,000, in contrast to the more than 700,000

MS² entries for the more than 30,000 compounds covered in the open libraries GNPS[12], MoNA and MassBank EU[13]. The mzCloud database contains the most extensive MSⁿ collection, with more than 16 million spectra for over 30,000 unique compounds (April 2024). Like other proprietary MS libraries, including Wiley, NIST23 and METLIN[14], data in mzCloud are locked and cannot be downloaded in open readable formats, thereby limiting their use in other tools and the training of machine learning models.

Here we introduce MSⁿLib, an open large-scale MSⁿ library that covers more than 30,000 unique compound structures in over 2.3 million MSⁿ spectra, and which significantly expands public MS resources. We describe our scalable, high-throughput pipeline combining collaborative compound sourcing, an open-source metadata curation script, rapid data acquisition, and automated data processing implemented in the open-source software mzmine[15]. Large-scale

[1]Institute of Organic Chemistry and Biochemistry of the Czech Academy of Sciences, Prague, Czechia. [2]Department of Pharmaceutical Sciences, Faculty of Life Sciences, University of Vienna, Vienna, Austria. [3]Skaggs School of Pharmacy and Pharmaceutical Sciences, University of California San Diego, San Diego, CA, USA. [4]Institute of Inorganic and Analytical Chemistry, University of Münster, Münster, Germany. [5]Institute of Microbiology of the Czech Academy of Sciences, Prague, Czechia. [6]Department of Biochemistry, University of California Riverside, Riverside, CA, USA. [7]Université Côte d'Azur, CNRS, ICN, Nice, France. [8]Interdisciplinary Institute for Artificial Intelligence (3iA) Côte d'Azur, Sophia Antipolis, Valbonne, France. [9]Department of Cell Biology, Faculty of Science, Charles University, Prague, Czechia. [10]These authors contributed equally: Corinna Brungs, Robin Schmid. ✉e-mail: rschmid1789@gmail.com; tomas.pluskal@uochb.cas.cz

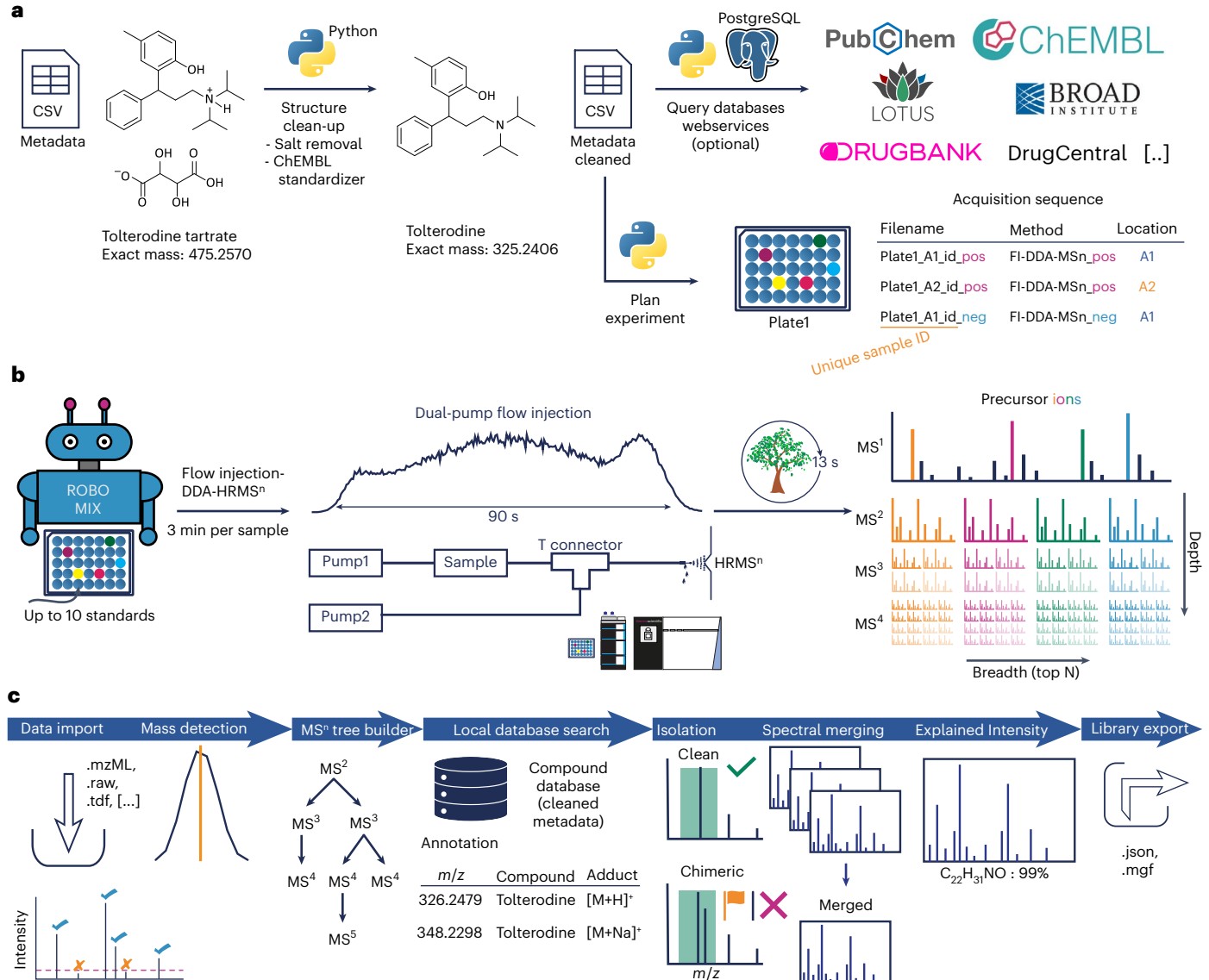

**Fig. 1 | Mass spectral library-building workflow. a**, Metadata clean-up and acquisition sequence generation. The metadata clean-up process consists of structure harmonization, experiment planning, and optional database queries to retrieve general compound, drug, natural product or other information. **b**, Sample preparation and flow injection data-dependent acquisition. The high-throughput data acquisition uses robotic and Echo liquid handling to mix and dilute compounds for subsequent analysis with a flow injection data-dependent acquisition $MS^n$ method. **c**, Data processing in mzmine. The automatic library generation workflow is implemented in mzmine and incorporates support for various data formats, processing steps, compound annotation, spectral merging, quality checks and export to open library formats. DDA, data-dependent acquisition; HRMS, high-resolution mass spectrometry.

spectral library efforts are challenged by limited access to diverse, often costly compound collections. Through a collaborative network, we obtained seven collections totaling 37,829 compounds and 34,413 unique structures, representing a broad chemical space across various natural and synthetic classes (Extended Data Table 1, Extended Data Fig. 1, Supplementary Note 1, Supplementary Fig. 1 and Supplementary File 1).

Our library generation pipeline consists of three stages: metadata curation, data acquisition, and data processing (Fig. 1). The main steps include metadata clean-up and acquisition sequence creation (Fig. 1a), followed by high-throughput data acquisition (Fig. 1b), and conclude with automated spectral library generation in mzmine (Fig. 1c). High-quality metadata curation is critical. We developed a Python script to clean input structures (SMILES/InChI), removing salts and applying harmonization steps (Fig. 1a). This script also enriches compound metadata by querying numerous chemical, biological and

drug databases and applying compound classifiers. Finally, unique sample identifiers (IDs) are assigned to aid in later data processing. Task orchestration via the Python-based Prefect framework manages and accelerates database queries (Supplementary Fig. 2). More details are given in Methods and Supplementary Note 2.

$MS^n$ data acquisition is challenging due to the long MS cycle times that are required to acquire deep and wide spectral trees (Extended Data Fig. 2). In a pilot study (Supplementary Note 3), we developed a high-throughput dual-pump flow injection method to capture high-quality $MS^n$ spectra for multiple adducts of up to 10 compounds per injection (Fig. 1b). Specifically the automatic gain control, injection time and mass resolution were optimized to raise spectral quality and signal-to-noise ratios (Extended Data Fig. 3 and Supplementary Fig. 3). Reproducible static noise signals were identified and later removed during data processing (Supplementary Fig. 4). Using the optimized method, all compounds were analyzed in both ionization modes in

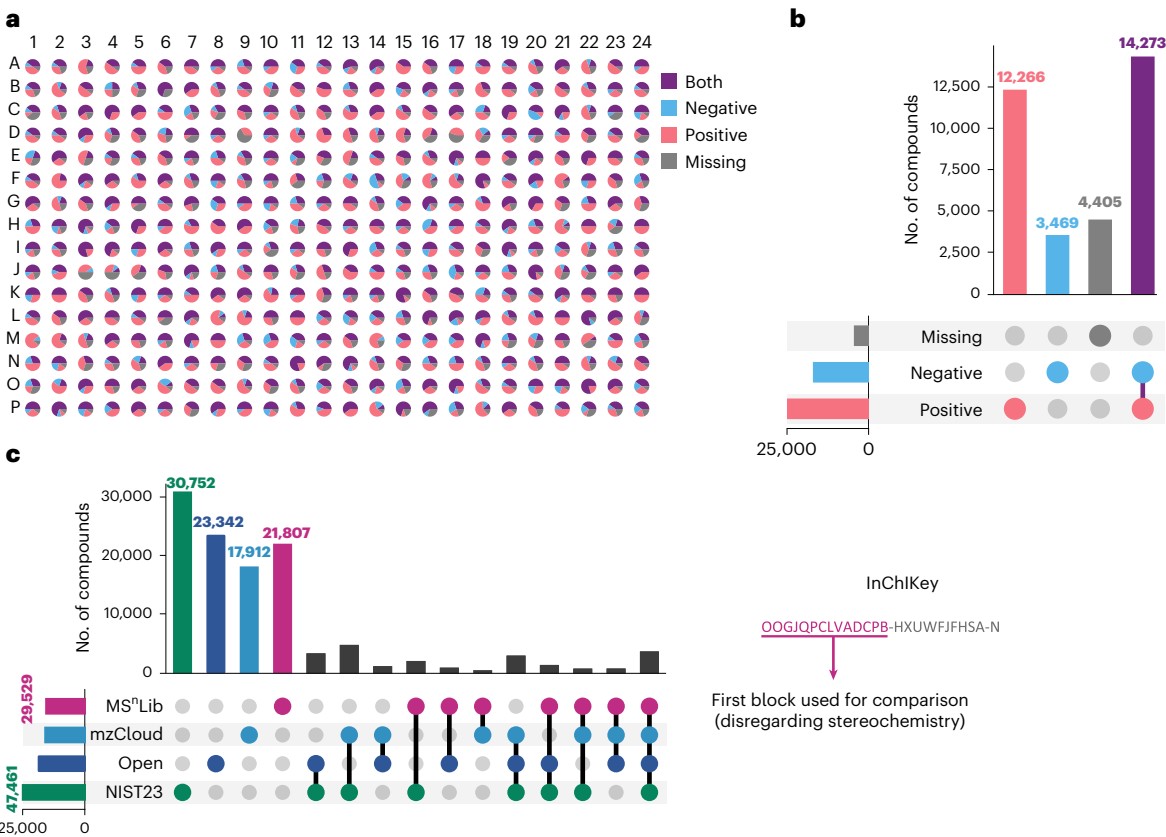

**Fig. 2 | Comparison of compounds in spectral libraries. a**, Plate visualization. A 384-well plate of the MCEBIO library is shown with wells as pie charts, depending on the percentage of compounds detected in each ionization mode. Ten compounds were mixed in each well. **b**, Detection dependent on ionization mode. The results for both polarities and for all seven libraries including 30,008 extracted unique compounds (with stereochemistry) are shown in an UpSet plot. **c**, Compound comparison with open and commercial spectral libraries. This UpSet plot shows the uniqueness and overlap of compounds of the newly acquired MS$^n$Lib compared with open (that is, GNPS (ALL_GNPS_NO_PROPAGATED), MassBank EU (MASSBANK_NIST) and MoNA (LC-MS/MS spectra), data from December 2023) and two commercial spectral libraries (mzCloud (auto and reference export), data from 4 March 2024; and NIST23 data from 13 August 2024) depending on the first InChIKey block, omitting stereochemistry.

23 days and 9,060 injections using auto-generated sequences with unique sample IDs.

Addressing the otherwise laborious data processing, we implemented an automated library-building workflow in mzmine (Fig. 1c). It imports data, builds MS$^n$ trees and annotates features against curated metadata by searching for compounds as various expected adducts or in-source fragments using their exact $m/z$ ratio. The sample's known composition constrained annotations. The mzmine feature table facilitates manual validation with annotations and ion traces (Supplementary Fig. 5). Key automated quality checks include precursor purity and fragment annotation rates. Finally, spectra were merged on different levels and exported in open MS library formats (Methods, Supplementary Note 4 and Supplementary Fig. 6).

Application of our automated workflow to the seven compound libraries successfully generated MS$^n$ trees for 30,008 unique compounds (87% coverage), yielding 357,065 MS$^2$ and over 2.3 million MS$^n$ spectra after merging and deduplication (Extended Data Tables 2 and 3). This establishes MS$^n$Lib as a large-scale open library and enables it to significantly expand publicly available MS$^n$ data. Achieving this high coverage required the combining of both ionization modes, given that many compounds were detected exclusively in positive (>12,200) or negative (>3,400) ionization, alongside those found in both (~14,300) (Fig. 2a,b, Supplementary File 1 and Supplementary Figs. 7–13). Comparison with existing libraries confirms MS$^n$Lib's complementarity, contributing 22,700 new compounds and valuable MS$^n$ trees (Fig. 2c). The unique chemical space covered by MS$^n$Lib and

projected using the data visualization method, TMAP[16] (Extended Data Fig. 4), underscores our method's capability to substantially expand spectral knowledge. Supplementary Fig. 14 shows a representative MS$^n$Lib entry in mzmine's MS$^n$ tree visualizer. Supplementary Note 5 provides a potential explanation for missed compounds.

The MS$^2$ spectral quality of the MCEBIO library entries, part of the greater MS$^n$Lib, was evaluated using feature-based molecular networking[17]. The pair-wise fragmentation similarity mapped the chemical space and led to the clustering of similar compound classes, proving high spectral quality. Only 20% of the entries matched to other open databases, highlighting the novelty of the MS$^n$Lib spectra (Extended Data Fig. 5 and Supplementary Note 6). For evaluation, we matched MS$^n$Lib against a public dataset of drug-incubated bacterial cultures (MSV000096589). While other combined open libraries yielded 129 annotations, MS$^n$Lib provided 80. Crucially, MS$^n$Lib contributed 21 unique annotations, increasing the total annotated features to 150. Of these unique MS$^n$Lib matches, 14 corresponded to added drugs, and the remaining seven were potential microbial metabolites (for example, ufiprazole from omeprazole) that were not further investigated. Detailed matching results and spectral plots are provided in Supplementary Files 10 and 11.

We introduce MS$^n$Lib, a large-scale, open MS$^n$ spectral library featuring >2.3 million MS$^n$ and >357,000 MS$^2$ spectra for 30,008 unique compound structures detected across both ionization modes. This resource was built with collaborative compound sourcing, a high-throughput MS$^n$ acquisition method, and an automated,

open-source processing workflow in mzmine[15]. By contributing ~22,700 complementary compounds to existing libraries, MS$^n$Lib significantly expands public spectral knowledge. We anticipate that this unique resource will greatly enhance untargeted liquid chromatography–mass spectrometry annotation and fuel machine learning advancements that utilize detailed MS$^n$ fragmentation and encoded substructure information for structure prediction and chemical classification. We aim to expand our collaborative network for compound and data sharing to further increase the coverage and impact of MS$^n$Lib. The automated library-building workflow is extendible and allows for efficient regeneration of MS libraries.

## Online content

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

## Methods

### Materials

All solvents used were of LC–MS grade. Methanol, acetonitrile, water and formic acid were obtained from Thermo Scientific. Seven different compound libraries, that were available to us, were analyzed: the NIH NPAC ACONN collection of natural products provided by the US National Institute of Health (NIH) with 3,988 compounds (NIHNP), a peptidomimetic library provided by OTAVAchemicals (Ontario, Canada) comprising 1,298 compounds (OTAVAPEP), the Discovery Diverse Set DDS-10 library from Enamine (Kyiv, Ukraine) with 10,240 compounds (ENAMDISC), a library mixture of 4,378 compounds purchased from Enamine and Molport (Riga, Latvia) including the 4,000 carboxylic acid fragment library (ENAMMOL), and three other libraries purchased from MedChemExpress (MCE, New Jersey, USA) containing 10,315 bioactive compounds (MCEBIO), 5,000 compounds from the MCE 5K Scaffold Library (MCESCAF), and 2,610 Food and Drug Administration-approved drugs (MCEDRUG). More information on compounds is given in Supplementary Note 1.

### General workflow

1. Metadata clean-up
   a. Manual harmonization of input metadata sheet and column names
   b. Defining jobs and running jobs.py
2. Sequence generation
3. Sample preparation (mixing compounds)
4. Data acquisition
5. Automatic $MS^n$ tree library creation and data evaluation in mzmine
   a. Combining the cleaned metadata with the acquired data

### Metadata clean-up

A Python script (metadata_cleanup_prefect.py) was developed for the curation of metadata. The main purpose of this script is structure extraction, salt removal, and standardization. Structure standardization was based on the ChEMBL structure pipeline Python package[18]. The complete clean-up failed with the original pipeline for specific structures (when a salt was given without a dot in the structure) and required an additional initial salt removal step and a second clean-up run. The cleaned and standardized structure is used for calculating other structural information such as canonical and, if available, isomeric SMILES (simplified molecular input line entry system), InChI (international chemical identifier developed by the International Union of Pure and Applied Chemistry), InChIKey (a condensed version of InChI), logP (the logarithm of the octanol–water partition coefficient) and monoisotopic mass. This mass is used during the automatic library generation. Optionally, additional information, such as whether a compound is considered a natural product or used as a drug, can be gathered from other databases based on a name, database identifier or structure search in PubChem, ChEMBL or other public resources (Supplementary Note 2). Queries can be easily turned off or implemented in the Python code. In our code, all databases that require a local file are deactivated by default.

### Sequence generation

An additional Python script (sequence_creation.py) was used to prepare the sequence table based on plate and well information. The sequence for each plate is built so that it is first analyzed in positive ion mode, followed by negative ionization. A file name is generated automatically and contains the date, a unique sample identifier (combination of library, plate number and well location), the method used, and the polarity. The unique sample identifier is important because it is used for the automatic annotation and extraction of the acquired spectral data. Therefore, the unique sample identifier in the metadata column needs to be matched with a substring of the acquisition file name. Here, only this identifier is important in the name, enabling the addition of other prefixes and suffixes, for example, the date, polarity or method. The script generates acquisition sequence files specific to the Xcalibur sequence layout using the flow injection–Orbitrap $MS^n$ method. This can be easily modified for other analysis platforms, depending on their layout.

### Sample preparation

Our mass spectral library contains seven different compound libraries. The MCEBIO library was prepared with an OT-2 liquid handler (Opentrons Labwork) to pool and dilute 10 compounds in each well of three 384-well plates, resulting in a concentration of 20 µM for each compound in a mixture of methanol and water (1:1). For the MCESCAF, MCEDRUG, OTAVAPEP, ENAMMOL and ENAMDISC libraries, the Echo 650 Liquid Handler (Beckmann Coulter) was used to pool eight compounds in 384-well plates, and a CERTUS FLEX liquid dispenser (Fritz Gyger AG) diluted the samples with 80–90 µl methanol and water (mixed in a 1:1 ratio), resulting in a concentration between 8 and 12 µM. The NIHNP library was further processed at the University of California San Diego, California, USA. Up to seven compounds were pooled in 96-well plates, resulting in a concentration of 5 µM. Because the plates showed strong evaporation, they were refilled with 50–100 µl methanol, acetonitrile and water (mixed in a 4:4:2 ratio), which was the previous mixture. Therefore, the end concentration is unknown.

### Data acquisition

The flow injection–$MS^n$ analysis was performed using a Vanquish Horizon UHPLC system with two pumps coupled to an Orbitrap ID-X (Thermo Fisher Scientific) instrument. The instrument was calibrated in positive and negative ion mode with the Pierce FlexMix calibration solution prior to a library batch. Different set-ups were tested to extend the peak width for more $MS^n$ experiments, to reach the mass analyzer quickly, and to avoid sample carryover. Two pumps were connected with a T-piece. The first pump, that is, the delivery pump, ran through the autosampler. The flow for this pump was initially set to 45 µl min⁻¹ for the injection to reach the T connection quickly. After 0.27 min the flow was decreased to 5 µl min⁻¹ and kept constant until 1.35 min. Over the next 0.15 min the flow was increased to 45 µl min⁻¹ and kept constant for another 1.5 min to clean the sample lines and to avoid sample carryover. The second pump, that is, the make-up pump, was used to broaden the elution profile. To maintain a combined constant flow of 55 µl min⁻¹, the make-up pump started at 10 µl min⁻¹, was increased to 50 µl min⁻¹ after 0.27 min, and kept at that rate until 1.35 min. The flow was gradually decreased back to 10 µl min⁻¹ over the next 0.15 min and kept constant for another 1.5 min. The whole run time per injection was 3 min. Both pumps used an isocratic mixture of water and acetonitrile at a 50:50 ratio, both with 0.1% formic acid. The switching of the flow speed is important for cleaning because the delivery pump runs at a low flow rate of 5 µl min⁻¹ during most of the time of the sample delivery. It must be noted that the method can also be used with a second pump running constantly at 50 µl min⁻¹, resulting in an altered flow rate of up to 95 µl min⁻¹ during the analysis but with no big changes during the data acquisition.

The injection volume was set to 2 µl, except for that for the NIHNP library, which was set to 3 µl due to the lower concentration. H-ESI was used for ionization with a vaporizer temperature of 75 °C and ion transfer tube temperature of 275 °C. The voltages were set to 3,000 V and 2,000 V for positive and negative ionization modes, respectively. The sheath gas was set to 25 a.u. and the auxiliary gas was set to 5 a.u. No sweep gas was used. The $MS^n$ tree was built with the following main settings, with the Orbitrap as the mass analyzer: For $MS^1$, data were analyzed from $m/z$ 115 to 2,000 with a resolution of 30,000, a radio-frequency (RF) lens of 50%, an automatic gain control (AGC) target of

100% (40,000 a.u.), and a maximum injection time (maxIT) of 50 ms. After one $MS^1$ scan, the three most intense ions, with a minimum intensity of $6 × 10^5$ a.u. in positive and $2 × 10^5$ a.u. in negative ionization, were picked using data-dependent acquisition with an isolation window of $m/z$ 1.2, a resolution of 15,000, an AGC target of 30% ($1.2 × 10^4$ a.u.), and maxIT of 50 ms for positive and 80 ms for negative ion mode. Three fragmentation experiments to cover different collision energies (a maximum of nine scans) were conducted. For $MS^2$, the energies were set to 20 eV and 60 eV, and the assisted collision energy to achieve the optimal $MS^2$ energy for further $MS^n$ stages was tested in 15 eV steps, starting at 15 eV, and increasing to 30 eV, 45 eV, 60 eV and 75 eV. From this assisted collision energy step, the top five signals, with a minimum intensity of $2 × 10^4$ a.u. in positive ion mode and $1 × 10^4$ a.u. in negative ion mode, and within the mass range of $m/z$ 90–2,000, were isolated for $MS^3$ with an $MS^1$ isolation window of $m/z$ 1.2 and an $MS^2$ isolation window of 2. The resolution was set to 60,000, the AGC target to 100% ($5 × 10^4$), and the maxIT to 200 ms for the positive and 500 ms for the negative ion mode. Three fixed collision energies of 20 eV, 40 eV and 60 eV were applied, resulting in a maximum number of 15 scans. The two most intense signals with a minimum intensity of $2 × 10^4$ a.u. in positive and $1 × 10^4$ in negative ion mode were selected from the 40 eV $MS^3$ scans for the $MS^4$ experiments, with an isolation window of $m/z$ 2.2. All other settings were used as in $MS^3$. For $MS^5$, the two highest signals of an 40 eV $MS^4$ scan and within a mass range of 150–2,000 were further fragmented using an isolation window of $m/z$ 3 and the same settings as $MS^3$ and $MS^4$. Only 40 eV and 60 eV were used as the collision energy settings, given that 20 eV produced mainly the precursor ion. Dynamic exclusion was carried out at every $MS^n$ stage, meaning that each precursor was selected three times within 200 s and was excluded for the following 70 s with a mass tolerance of $m/z$ 0.2. Additionally, isotopes of selected precursor ions were excluded within a window of $m/z$ 2 for unassigned isotopes. It must be noted that the maximum occurrence should be set to a number divisible by 3 to carry out experiments for all three collision energies. Before the standard analysis, multiple blank injections were analyzed, and the detected signals were added to a targeted mass exclusion list. This exclusion list was updated after running 10 sample injections and the detection of reoccurring signals in all samples. The $MS^n$ schema is presented in Extended Data Fig. 2 and for one example compound in Supplementary Fig. 14. The processing was done in mzmine. A full batch configuration file is supplied as Supplementary File 2 (mzmine_exclusion_blankprofile_pos.mzbatch) and Supplementary File 3 (mzmine_exclusion_blankprofile_neg.mzbatch). Our system showed higher background signals in empty scans or less rich fragmentation scans around $m/z$ 149.72 and $m/z$ 173.52, therefore, these were added to an exclusion list with a width of $m/z$ 0.03 for all $MS^n$ levels. The fragmentation tree is presented in Extended Data Fig. 5 with an example given in Supplementary Fig. 14. All settings are listed in Supplementary Tables 1 and 2.

### Automatic $MS^n$ tree library generation and data evaluation in mzmine

The automatic library generation workflow was implemented in mzmine, to provide support for MS data from various vendors and open formats, spectral processing, spectral quality assessment, and annotation based on curated metadata. A spectral library generation workflow and a flow injection workflow were added to the mzwizard module, which is embedded in mzmine. The mzwizard supports a simplified workflow set-up while still preserving full configurability of the final workflow.

The data processing and automatic library extraction were done in mzmine using the steps below. A full batch configuration file is supplied as Supplementary File 4 (mzmine_msn_library_pos.mzbatch) and Supplementary File 5 (mzmine_msn_library_neg.mzbatch), and the mzwizard configuration is provided as Supplementary File 6 (mzwizard_msn_library.mzmwizard).

1. Import of Orbitrap MS data as .raw files or as .mzML files after conversion using the ThermoRawFileParser (https://github.com/compomics/ThermoRawFileParser) or the MSConvert script (https://proteowizard.sourceforge.io/download.html).
2. MS data processing
   a. Denoising: mass detection on $MS^n$ with the factor of the lowest signal mass detector and noise factor of 2.5 for all MS levels
   b. Background signal removal of two known artifacts
   c. Tree building
   d. Compound annotation based on a local compound database search. Here, the monoisotopic mass is used together with various selected ion adducts and in-source fragments to calculate the precursor mass. The algorithm considers only compounds in specific samples matched by a unique sample ID substring in the file names.
3. Spectral library export to .json, .mgf or .msp formats
   e. Scoring of the precursor isolation purity (%) of $MS^n$ spectra based on the preceding and following $MS^1$ scan. Chimeric spectra are flagged in the output file.
   i. Export the best spectrum for each precursor and energy (highest total ion chromatogram), no SPECTYPE information or 'SINGLE_BEST_SCAN' in library file
   f. Merging of spectra, SPECTYPE information in the library file:
   i. 'SAME_ENERGY': each individual fragmentation energy, when triggered multiple times in the same sample
   ii. 'ALL_ENERGIES': all fragmentation energies for individual precursors (three energies in our method, using the merged same_energy if available, otherwise the best one)
   iii. 'ALL_MSN_TO_PSEUDO_MS²': combining the full $MS^n$ tree of a compound ion into a pseudo-$MS^2$ spectrum
   g. Filtering of spectra based on a minimum of two signals above the noise threshold
4. (Optional) Reimport of the spectral library to check the success of its generation
5. (Optional) Alignment of all feature lists across samples and their matching to the newly generated spectral library as initial validation
6. (Optional) Manual inspection of the spectral libraries and $MS^n$ experiments using the $MS^n$ tree visualizer (see Supplementary Fig. 15 for an example).

The processing was performed for 11,000 compounds in 1,100 injections on a DELL XPS 15 9510 laptop with 32 GB of RAM, eight processor cores and 16 threads for speed testing of the automatic library generation.

Various information can be stored within the library file, including retention time, ion mobility, collision energy, as well as instrument, method or compound specifications.

### Compound list comparison between $MS^n$Lib and other spectral databases (TMAP projection)

Prior to the comparison, we cleaned and standardized the structure of all resources in the same way and calculated the InChIKey. The comparison is based on the first InChIKey block, removing stereochemistry. The libraries used are included in the Data Availability section.

### Feature-based molecular networking and matching against public mass spectral libraries

The newly generated spectral library in positive ion mode for the MCE-BIO library was imported into mzmine and reprocessed to a feature list. Only $MS^2$ and pseudo-$MS^2$ spectra were used, resulting in 48,069 spectra, to reduce data complexity, and given that most tools are limited to use with $MS^2$ spectra. The feature list annotation was exported as a .csv file retaining the original information, for example, compound name

and adduct for later comparison. Furthermore, the list was exported with the mzmine module named molecular networking files in an .mgf data format, compatible with running feature-based molecular networking (FBMN) utilizing the Global Natural Products Social Molecular Networking (GNPS) infrastructure. The parameters for FBMN were set to precursor ion and fragment ion mass tolerances of 0.02 Da, a minimum pairs cosine value (min. pairs cos.) of 0.7 (minimum cosine score necessary to connect to experimental MS$^2$ spectra), a network TopK value of 1,000, a minimum number of matched fragment ions of 4, maximum connected component size of 0, and a maximum shift between precursors of 500 Da.

For matching our library against the three most commonly used public spectral databases, we used LC−MS/MS spectra from the MassBank of North America (MoNA, https://mona.fiehnlab.ucdavis.edu/downloads,.json, accessed 8 December 2023), spectra from the MassBank EU database, specifically MassBank_NIST.msp (https://github.com/MassBank/MassBank-data/releases/tag/2023.11), and spectra from the GNPS library, namely ALL_GNPS_NO_PROPOGATED (https://gnps-external.ucsd.edu/gnpslibrary, accessed 8 December 2023). Prior to matching our MS$^n$Lib's feature list against these public spectral databases (see the first step), the original annotations were removed to retain only spectral library matches. We used the following settings: a minimum matched signals setting of 4, a precursor $m/z$ tolerance and spectral $m/z$ tolerance of 0.005 or 10 ppm, removing the precursor, a weighted cosine similarity with a minimum similarity of 0.6, and the weighting of the square root of the signal intensity ($m/z^0 \times I^{0.5}$). The top five and top 10 matches were exported for further evaluation. Here, the feature ID produced by mzmine was used to compare the annotations by spectral matching with the original compound information. The structures of the matched compounds were cleaned with the same script as in the metadata clean-up and a new InChIKey string was computed. This InChIKey string was used to find spectra that were matched to the identical compound. Finally, to further evaluate the top 10 matching hits, we calculated their Tanimoto similarity, based on Morgan fingerprints (radius = 2, nBits = 2048), and determined their maximum common edge subgraph (MCES)[19], using the default settings in the Python package. The FBMN visualization is provided in Supplementary File 7.

**Reporting summary**

Further information on research design is available in the Nature Portfolio Reporting Summary linked to this article.

## Data availability

All metadata files, one per compound library, and mzmine batch files for library processing were uploaded to the MERLIN (Mass spEctRal LIbrary Network) GitHub repository (https://github.com/merlin-ms) under the MIT license. All acquired flow injection−Orbitrap MS$^n$ files were deposited as .mzML and .raw files in the Zenodo datasets: 10966280 (.mzML positive and negative), 10966404 (.raw positive) and 10967081 (.raw negative) under the CC BY 4.0 license, and were uploaded to MassIVE MSV000094528 under the CC0 1.0 license. The mass spectral libraries included in MS$^n$Lib were deposited as .mgf and .json files in the Zenodo dataset 11163380 under the CC BY 4.0 license. Here, each spectral library for the individual compound libraries is uploaded as MS$^2$ only or the full MS$^n$ library. The MS$^2$ libraries contain the best and merged spectra for all acquired MS$^2$ spectra, including the pseudo-MS$^2$, in which the whole fragmentation tree is combined into a single spectrum. Polarities are kept separated, resulting in four entries for each compound library and library format. Additionally, the MS$^2$ data are uploaded as a reference library in GNPS (https://external.gnps2.org/gnpslibrary) in the form of a default gold-level library named MSNLIB-POSITIVE and MSNLIB-NEGATIVE. We recommend using the Zenodo libraries because they contain more metadata and link back to the original data by universal spectrum identifier (USI). Regarding

the metadata clean-up, the DrugBank lookup is an optional step, and the data are accessible on request. More information is available on the project website (https://go.drugbank.com). DrugCentral lookup is an optional step, and their whole database can be downloaded as a PostgreSQL dump from the company's website (https://drugcentral.org). LOTUS lookup is an optional step, and the whole LOTUS dataset is incorporated into WIKIDATA. The metadata clean-up script contains a Prefect flow to download all relationships from WIKIDATA. Simply run the prepare_wikidata_lotus_data_prefect.py script (https://github.com/corinnabrungs/msn_tree_library). Broad Institute lookup is an optional step. The drug information can be downloaded as a .txt file from the institute's website (https://repo-hub.broadinstitute.org/repurposing). The FBMN result can be accessed on GNPS: https://gnps.ucsd.edu/ProteoSAFe/status.jsp?task=c05d34fb31ab4ee99293e722fb7eb83d Experimental public libraries were downloaded from the corresponding webpages: GNPS, ALL_GNPS_NO_PROPAGATED (https://external.gnps2.org/gnpslibrary, accessed 8 December 2023); MassBank EU, MASSBANK_NIST (https://github.com/MassBank/MassBank-data/releases/tag/2023.11, accessed 8 December 2023); MassBank North America, LC−MS/MS spectra (https://mona.fiehnlab.ucdavis.edu/downloads, accessed 8 December 2023). The dataset used for the library evaluation can be found in MassIVE under MSV000096589. Here we used only a subset of group 1 (group1_B*.mzML) for the analysis. The mzmine batch file is supplied as Supplementary File 9 and the results of this evaluation as Supplementary Files 10 and 11.

## Code availability

The metadata clean-up pipeline was implemented in Python 3.10, with the Prefect library as the task orchestration tool. The source code is available on GitHub under the free and open MIT license (https://github.com/corinnabrungs/msn_tree_library). The TMAP projections and other analysis are available as Jupyter notebooks on GitHub (https://github.com/corinnabrungs/msn_tree_library/tree/master/notebooks). The mzmine code and documentation are available on GitHub under the same license (https://github.com/mzmine/mzmine3, https://github.com/mzmine/mzmine_documentation).

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

## Acknowledgements

C.B. was supported by the Czech Academy of Sciences PPLZ fellowship number L200552251. The mzmine project is funded by the European Union, the BAB—funding bank for Bremen and Bremerhaven, and the Senator of Economics, Ports and Transformation Bremen (65002459). A.M. was supported by the project National Institute of Virology and Bacteriology (Programme EXCELES, ID Project Number LX22NPO5103)—funded by the European Union—Next Generation EU. P.C.D was supported by grants from the National Institute of Health R01DK136117 and R01GM107550. Z.K. was supported by the Ministry of Education, Youth and Sports of the Czech Republic grant CZ.02.01.01/00/22_008/0004597 within the One Health framework. T.P. was supported by the Czech Science Foundation (GA CR) grant 21-11563M and by the European Union's Horizon 2020 research and innovation programme under Marie Skłodowska-Curie grant agreement Number 891397. We acknowledge core facility structural mass spectrometry of the Czech Infrastructure for Integrative Structural Biology (CIISB), Instruct-CZ Centre, supported by the Ministry of Education, Youths and Sports of the Czech Republic (MEYS CR; LM2023042) and European Regional Development Fund-Project 'Innovation of Czech Infrastructure for Integrative

Structural Biology' (Number CZ.02.01.01/00/23_015/0008175). The funders had no role in study design, data collection and analysis, decision to publish or preparation of the manuscript. We thank O. Mokshyna and R. Bushuiev for fruitful discussions on metadata standardization and chemical space visualization. We thank M. Ulaszewska and T. Mak for providing information about mzCloud and NIST23, respectively, and acknowledge the support of the Dagstuhl computational metabolomics communities. The conceptual basis of this project was formed at Dagstuhl Seminars 20051 and 22181. We thank F. Rooks for editing this paper.

## Author contributions

C.B., R.S. and T.P. conceived the method and wrote the paper. D.P., L.-F.N., Z.K. and P.C.D. gave feedback on the method and results. C.B. and R.S. wrote the metadata clean-up Python script. C.B. performed MS data acquisition and spectral library extraction. R.S., S.H. and C.B. added new modules for the automatic library building in mzmine. Z.K., V.V. and R.N. provided compound libraries. A.M., M.D., P.S. and L.-F.N. prepared the mixing and dilution of the compound libraries. M.D. developed a script to optimize the mixing of compounds, avoiding interferences. S.H., A.M., M.D., P.S., P.C.D., D.P., L.-F.N. and Z.K. provided feedback for the paper. All authors read and approved the paper.

## Competing interests

T.P., S.H. and R.S. are co-founders of mzio GmbH, which develops the mzmine software for mass spectrometry data processing. P.C.D. is an advisor and equity holder in the companies Cybele and Sirenas; a science advisor and equity holder in bileOmix; a scientific co-founder, advisor and equity holder of Ometa, Enveda and Arome, with prior approval by the University of California San Diego; and consulted for DSM Animal Health in 2023. The other authors declare no competing interests.

## Additional information

**Extended data** are available for this paper at https://doi.org/10.1038/s41592-025-02813-0.

**Correspondence and requests for materials** should be addressed to Robin Schmid or Tomáš Pluskal.

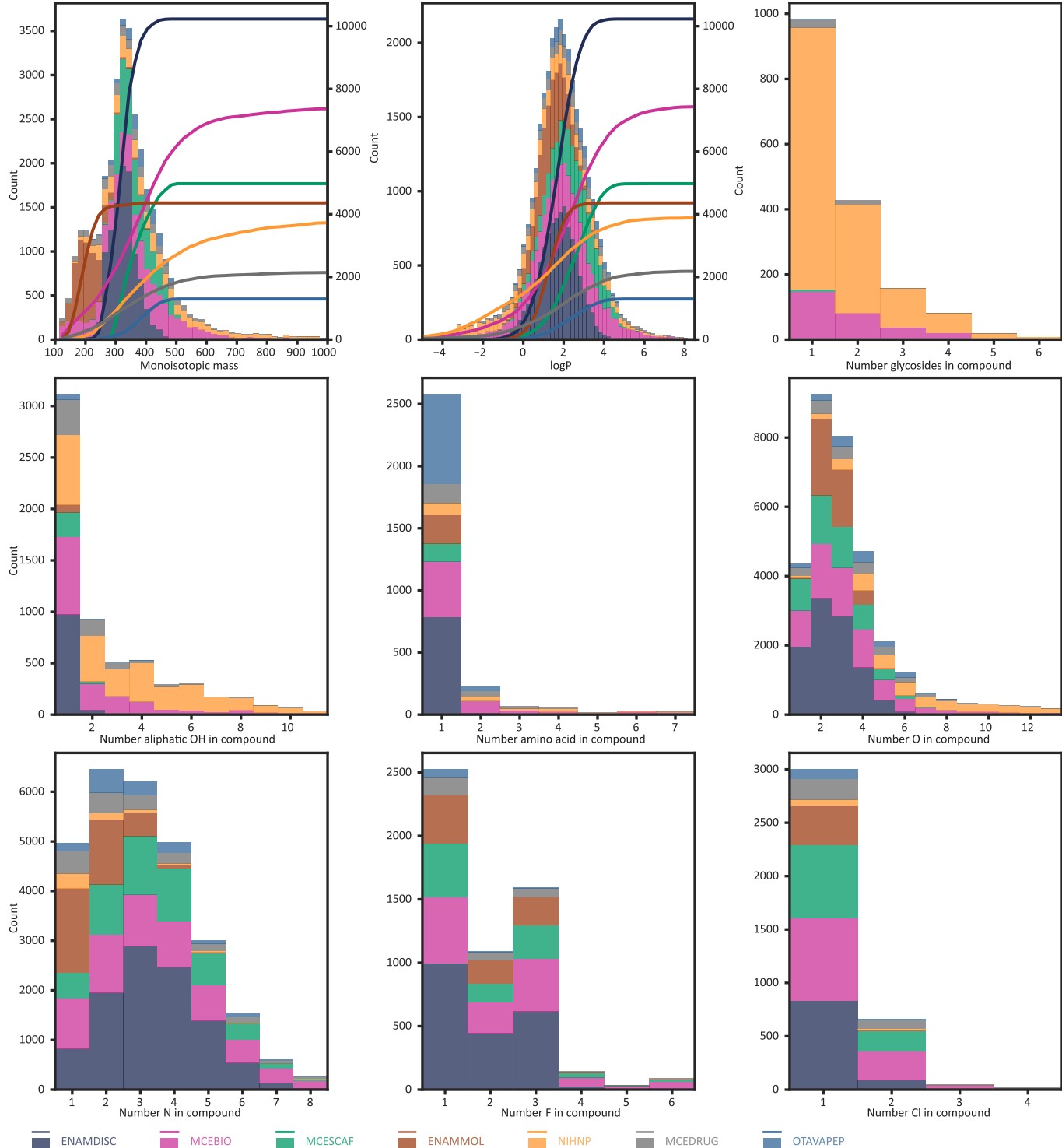

**Extended Data Fig. 1 | Insights into extracted chemical properties covered by the seve compound libraries ENAMDISC, MCEBIO, MCESCAF, ENAMMOL, NIHNP, MCEDRUG, and OTAVAPEP.** If a compound (the same InChIKey) is included in multiple compound libraries, it is kept in the smallest one.

Most compounds fall into the size range of small molecules with masses below 600 and logP values below 5, following Lipinski's rule. The compound libraries show differences in their covered compound classes and atomic compositions. Supplementary Fig. 1 presents additional histograms.

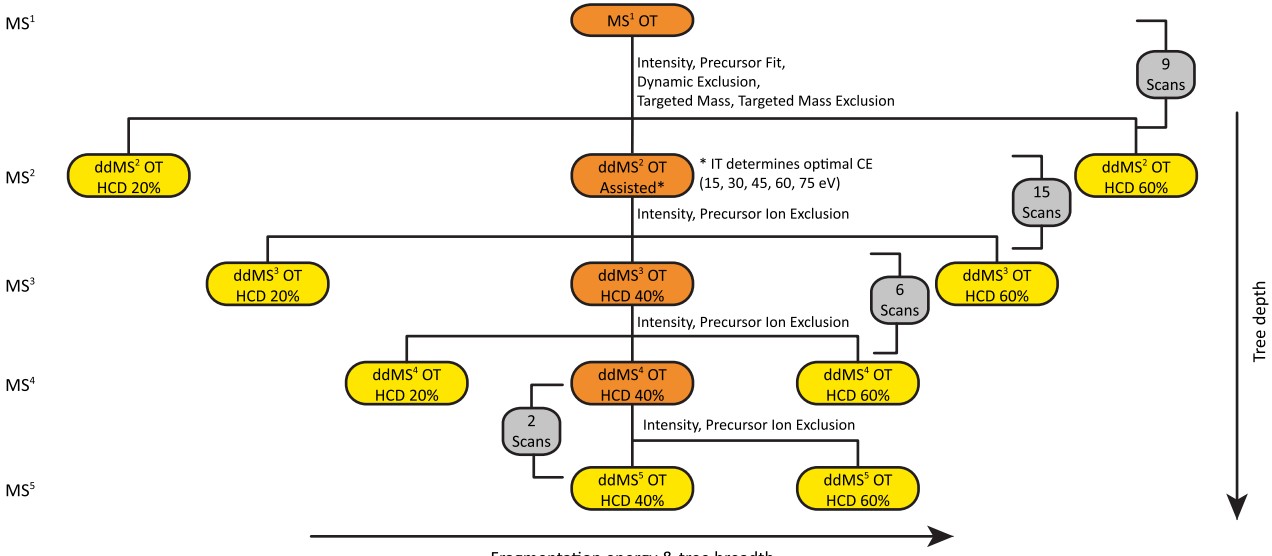

**Extended Data Fig. 2 | MSⁿ tree data acquisition.** Instrument method set-up that was used for the MSⁿ data acquisition. Three collision energies are applied to each precursor, resulting in Top5 for MS³, Top2 for MS⁴, and Top1 for MS⁵, resulting in a maximum number of 75 scans per MS² precursor ion. OT = Orbitrap; IT = Ion trap; dd = data-dependent, HCD = higher-energy collisional dissociation.

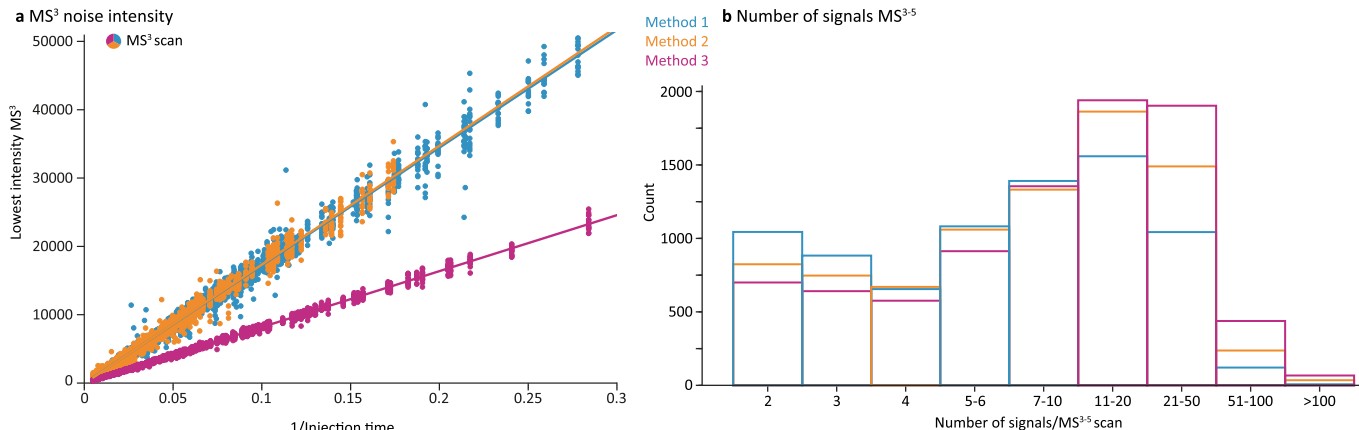

**Extended Data Fig. 3 | Comparison of three different Orbitrap methods influencing the noise intensity and number of signals per MS$^{3-5}$ scan for 220 pure standards in 22 samples of the MCEBIO library. a**, The noise intensity, here defined by the lowest signal in each MS$^3$ scan, is influenced by the injection time of the C trap. Internal processing by the instrument vendor normalizes the intensity values by dividing them by the injection time, leading to higher absolute noise levels for shorter injection times. A higher AGC target (Method 2) shifts to longer injection times, keeping the noise level on the same trajectory. A higher mass resolution decreases the noise level to a lower trajectory. **b**, The MS$^{3-5}$ scan quality is compared by evaluating the number of signals in all annotated and exported MS$^{3-5}$ library scans. This includes the noise removal of signals below 2.5 times the lowest signal intensity in each scan before spectral export. The method optimization increased the number of high-quality spectra with more than ten fragment signals and the overall number of exported library spectra, despite the lower scan rate. Method 1 (blue): AGC 60%, Resolution 15k; Method 2 (orange): AGC 100%, Resolution 15k; Method 3 (magenta): AGC 100%, Resolution 60k.

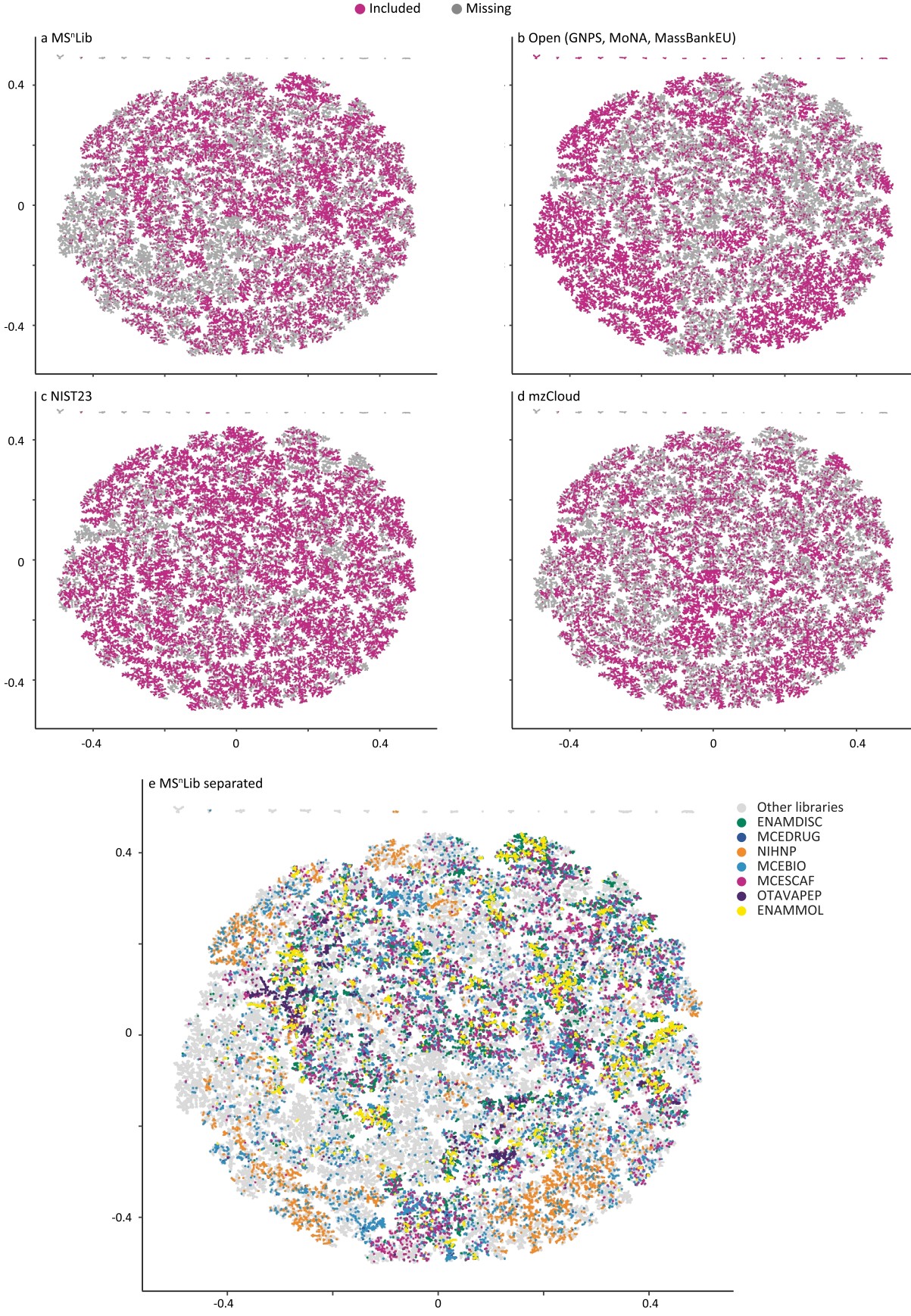

**Extended Data Fig. 4 | See next page for caption.**

**Extended Data Fig. 4 | TMAP projections of the chemical space coverage of MSⁿLib in comparison to other open spectral libraries and the commercial spectral libraries NIST23 and mzCloud.** The libraries are divided in **a** MSⁿLib, containing different compound libraries, **b** Open libraries, including GNPS, MoNA, and MassBank EU, (access date 08.12.2023, 35,278 unique compounds) **c** NIST23 (access date 13.08.2024, 47,461 unique compounds) and **d** mzCloud (access date 04.03.2024, 30,248 unique compounds). Each node represents an individual structure based on its canonical SMILES representation. Compounds included in the corresponding library are highlighted in magenta and drawn on top of compounds from other libraries in gray. The same structure can be included in multiple libraries. **e**, demonstrates a more fine-grained MSⁿLib coverage resolved by the individual compound libraries.

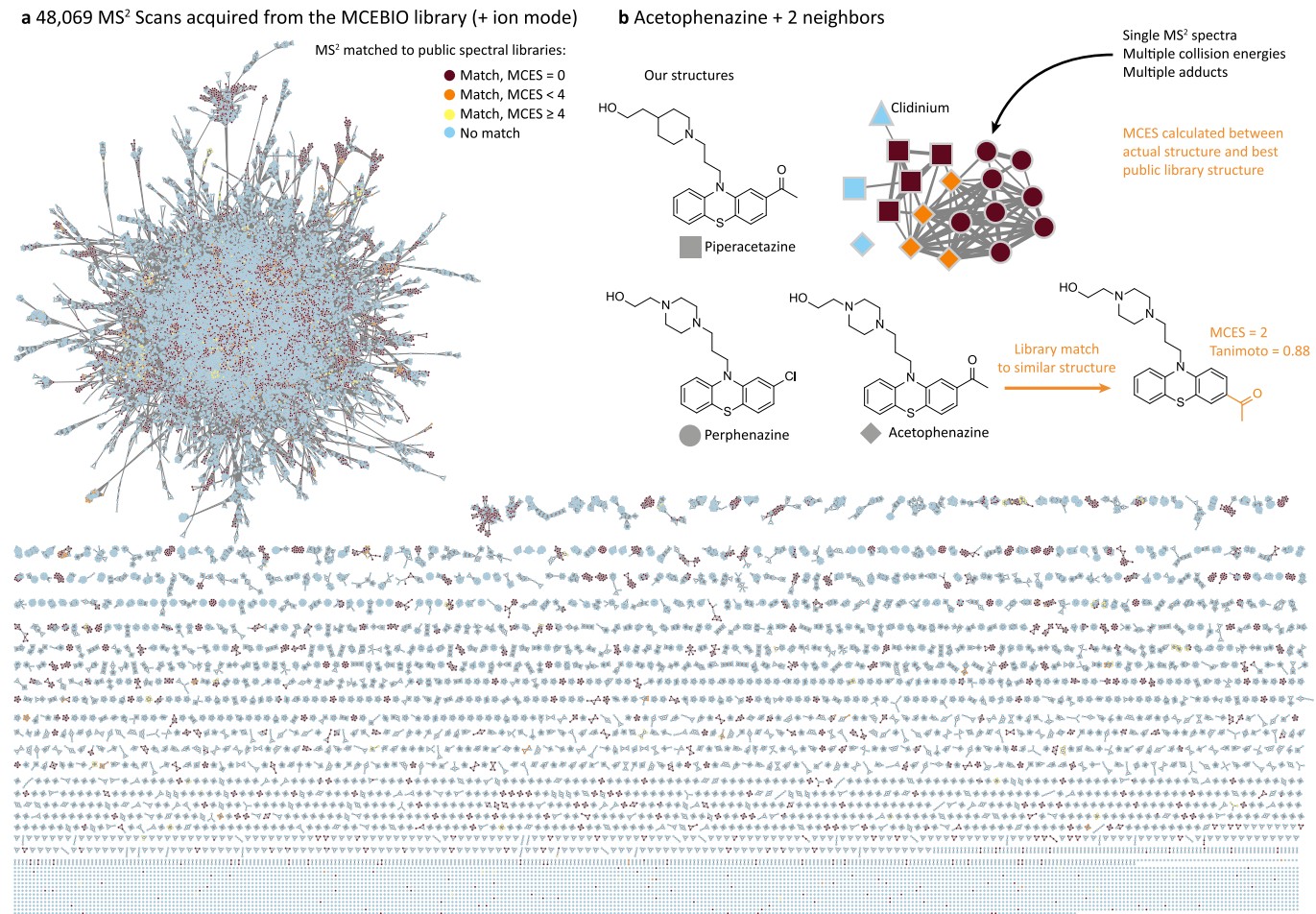

**a** 48,069 MS² Scans acquired from the MCEBIO library (+ ion mode)

MS² matched to public spectral libraries:

- ● Match, MCES = 0
- ● Match, MCES < 4
- ● Match, MCES ≥ 4
- ● No match

**b** Acetophenazine + 2 neighbors

Our structures

Piperacetazine

Perphenazine

Acetophenazine

Clidinium

Single MS² spectra
Multiple collision energies
Multiple adducts

MCES calculated between actual structure and best public library structure

Library match to similar structure

MCES = 2
Tanimoto = 0.88

**Extended Data Fig. 5 | Map of the chemical space covered by the MCEBIO MS² library in positive ion mode and matches to public spectral libraries.**
**a**, Each node represents a single MS² spectrum (MCEBIO) and the edges between two nodes show the spectral similarity of these two spectra (cosine similarity ≥ 0.7 and at least 4 matched signals). The clustering by feature-based molecular networking (FBMN) on the GNPS web platform highlights the coverage of the chemical space for the acquired 48,069 MS² MCEBIO spectra. The mzmine algorithm matched all spectra (nodes) against the open spectral libraries from GNPS, MoNA, and MassBank EU, with only a 20% annotation rate (colors). The top ten annotations for each scan were exported for structure clean-up and scoring of structural similarity with actual compounds analyzed. For this, the maximum common edge subgraph (MCES, https://github.com/AlBi-HHU/myopic-mces) and the Tanimoto similarity were calculated. MCES reflects the number of

structural modifications, attaining a value of 0 for identical structures. MCES < 4 was chosen for similar structures, and MCES ≥ 4 was selected for lower or no similarity. Of all annotations, 94% were identical or similar. **b**, A subnetwork for acetophenazine and its neighbors for a distance of up to two edges. The node shapes reflect actual structures, and colors reflect MCES scores. Spectral similarity successfully clustered this molecular family of analog structures. The spectral library match of acetophenazine to an analog demonstrates how MCES = 2 often describes structural isomers differing in the position of one functional group. Piperacetazine and perphenazine yielded identical matches. Clidinium belongs to a different chemical class. Overall, most spectra contain enough information to be connected by cosine similarity but remain unmatched against public libraries. The molecular network is shared as a Cytoscape file for interactive exploration (Supplementary File 7).

**Extended Data Table 1 | Metadata of compounds included for the new high-throughput library-building workflow**

| Library Description | | | | Compound Information | | NP Information | | | Clinical Information | |
|---|---|---|---|---|---|---|---|---|---|---|
| Short name | Full name | Company/ Provider | Catalog number | Total comp. | Unique InChIKeys (incl. stereochem.) | NPs flagged in ChEMBL | Present in LOTUS | Present in Dictionary of NPs | Any clinical phase | Clinical phase 4 |
| **MCEBIO** | Bioactive Compound Library +additional | MedChemExpress | HY-L001 | 10317 | 9680 | 3275 | 2420 | 1981 | 5243 | 2479 |
| **MCESCAF** | 5k Scaffold Library | MedChemExpress | HY-L902 | 4998 | 4998 | 43 | 10 | 8 | 54 | 36 |
| **NIHNP** | NIH NPAC ACONN collection of NPs | NIH/NCATS | - | 3988 | 3915 | 677 | 2476 | 2659 | 143 | 57 |
| **OTAVAPEP** | Alpha-helix Peptidomimetic Library | OTAVAchemicals | a-helix-Peptido | 1298 | 1297 | 42 | 2 | 0 | 3 | 0 |
| **ENAMDISC** | Discovery Diversity Set - 10 | Enamine | DDS-10 | 10.24 | 10240 | 2 | 2 | 3 | 1 | 0 |
| **ENAMMOL** | Carboxylic Acid Fragment Library + additional | Enamine and Molport | | 4378 | 4370 | 85 | 38 | 29 | 37 | 15 |
| **MCEDRUG** | FDA-Approved Drug Library | MedChemExpress | HY-L022 | 2610 | 2190 | 1376 | 522 | 332 | 2190 | 2190 |
| **Summary** | | | | 37829 | 34413 | 4003 | 4781 | 4510 | 5530 | 2691 |

Each compound's natural product information is extracted from ChEMBL, LOTUS or the Dictionary of Natural Products (Version 31.2). Clinical phases (1–4) are gathered from multiple public resources, including ChEMBL, DrugBank, DrugCentral and Broad Institute. Those queries are included in the Python script for metadata clean-up. NP, natural product.

## Extended Data Table 2 | Statistics on each library and acquisition mode

| Library | Ion mode | Runs | Total compounds (unique InChIKeys) | Detected compounds (unique InChIKeys) | | Scans MS2 | MS2 - MS5 | Per detected compound MS2 | MS2 - MS5 | Precursor purity | Mean signals/scan | Formula explains mean Intensity | Signals |
|---|---|---|---|---|---|---|---|---|---|---|---|---|---|
| MCEBIO | Both | - | 10,317 (9680) | 8289 (7788) | 80% (80%) | 85382 | 504776 | - | - | - | - | - | - |
| | Positive | 1051 | | 7058 (6628) | 68% (68%) | 48245 | 374412 | 7 | 53 | 98% | 21 | 98% | 94% |
| | Negative | 1051 | | 4843 (4579) | 47% (47%) | 37137 | 130364 | 8 | 27 | 98% | 12 | 97% | 90% |
| MCESCAF | Both | - | 4998 (4998) | 4378 (4378) | 88% (88%) | 42931 | 318737 | - | - | - | - | - | - |
| | Positive | 625 | | 4102 (4102) | 82% (82%) | 26283 | 243033 | 6 | 59 | 98% | 26 | 98% | 93% |
| | Negative | 625 | | 2143 (2143) | 43% (43%) | 16648 | 75704 | 8 | 35 | 98% | 14 | 98% | 91% |
| NIHNP | Both | - | 3988 (3915) | 3266 (3219) | 82% (82%) | 36338 | 208406 | - | - | - | - | - | - |
| | Positive | 530 | | 2330 (2305) | 58% (59%) | 19241 | 122837 | 8 | 53 | 98% | 20 | 98% | 95% |
| | Negative | 530 | | 2597 (2559) | 65% (65%) | 17097 | 85569 | 7 | 33 | 98% | 19 | 99% | 92% |
| OTAVAPEP | Both | - | 1298 (1297) | 1167 (1166) | 90% (90%) | 12739 | 95078 | - | - | - | - | - | - |
| | Positive | 163 | | 1070 (1069) | 82% (82%) | 6670 | 63123 | 6 | 59 | 99% | 24 | 99% | 93% |
| | Negative | 163 | | 922 (921) | 71% (71%) | 6069 | 31955 | 7 | 35 | 98% | 16 | 99% | 90% |
| ENAMDISC | Both | - | 10240 (10240) | 10032 (10032) | 98% (98%) | 116600 | 805195 | - | - | - | - | - | - |
| | Positive | 1280 | | 9545 (9545) | 93% (93%) | 68287 | 600566 | 7 | 63 | 98% | 20 | 98% | 93% |
| | Negative | 1280 | | 5431 (5431) | 53% (53%) | 48313 | 204629 | 9 | 38 | 98% | 14 | 97% | 88% |
| ENAMMOL | Both | - | 4378 (4370) | 3335 (3332) | 76% (76%) | 37177 | 253636 | - | - | - | - | - | - |
| | Positive | 550 | | 2704 (2701) | 62% (62%) | 24352 | 198866 | 9 | 74 | 97% | 21 | 95% | 86% |
| | Negative | 550 | | 1955 (1955) | 45% (45%) | 12825 | 54770 | 7 | 28 | 97% | 13 | 96% | 82% |
| MCEDRUG | Both | - | 2610 (2190) | 2267 (1891) | 87% (86%) | 26406 | 168362 | - | - | - | - | - | - |
| | Positive | 331 | | 2030 (1695) | 78% (77%) | 15268 | 123389 | 8 | 61 | 98% | 21 | 98% | 93% |
| | Negative | 331 | | 1284 (1083) | 49% (49%) | 11138 | 44973 | 9 | 35 | 99% | 15 | 97% | 90% |
| Summary | Positive | | | 28840 (26540) | 76% (77%) | 208346 | 1726226 | 7 | 60 | 98% | 22 | 98% | 92% |
| | Negative | | | 19173 (17742) | 51% (52%) | 149227 | 627964 | 8 | 33 | 98% | 15 | 98% | 89% |
| | **Both** | **9060** | **37829 (34413)** | **32732 (30008)** | **87% (87%)** | **357,573** | **2,354,190** | **8** | **47** | **98%** | **18** | **98%** | **91%** |

All 9,060 injections were measured in 23 days. The numbers of scans are containing the best and merged spectra including the pseudo-MS$^2$ spectrum, when a whole fragmentation tree is merged into a single spectrum. Detailed patterns of compounds detected in positive and negative ion modes are given in Supplementary Fig. 7–13.

**Extended Data Table 3 | Number of spectra for each MS level without merging (single best scan) and for individual and merged spectra (Spectype)**

| File | SINGLE_BEST_SCAN (experimental) | | | | Spectype (experimental or merged) | | | |
|---|---|---|---|---|---|---|---|---|
| | MS2 | MS3 | MS4 | MS5 | SINGLE_BEST_SCAN | SAME_ENERGY | ALL_ENERGIES | PSEUDO_MS2 |
| 20241003_enamdisc_neg_msn.mgf | 23805 | 64310 | 30228 | 537 | 118880 | 31665 | 46461 | 7623 |
| 20241003_enamdisc_pos_msn.mgf | 31463 | 152514 | 153508 | 8763 | 346248 | 115660 | 128516 | 10142 |
| 20241003_enammol_neg_msn.mgf | 6033 | 16262 | 7652 | 131 | 30078 | 11200 | 11488 | 2004 |
| 20241003_enammol_pos_msn.mgf | 11000 | 50658 | 43429 | 1617 | 106704 | 49771 | 38851 | 3540 |
| 20241003_mcebio_neg_msn.mgf | 18416 | 42277 | 14306 | 492 | 75491 | 17272 | 31268 | 6373 |
| 20241003_mcebio_pos_msn.mgf | 22874 | 93154 | 94145 | 13205 | 223378 | 57050 | 86206 | 7786 |
| 20241003_mcedrug_neg_msn.mgf | 4948 | 12944 | 5275 | 225 | 23392 | 10178 | 9602 | 1648 |
| 20241003_mcedrug_pos_msn.mgf | 6690 | 28696 | 26367 | 3670 | 65423 | 30458 | 25328 | 2180 |
| 20241003_mcescaf_neg_msn.mgf | 8253 | 22958 | 12275 | 674 | 44160 | 11455 | 17360 | 2729 |
| 20241003_mcescaf_pos_msn.mgf | 12530 | 58223 | 70364 | 8030 | 149147 | 34584 | 55108 | 4194 |
| 20241003_nihnp_neg_msn.mgf | 8412 | 26433 | 16829 | 1399 | 53073 | 8292 | 21363 | 2841 |
| 20241003_nihnp_pos_msn.mgf | 9285 | 34135 | 25043 | 2270 | 70733 | 21538 | 27485 | 3081 |
| 20241003_otavapep_neg_msn.mgf | 3163 | 11079 | 5764 | 330 | 20336 | 2525 | 7984 | 1082 |
| 20241003_otavapep_pos_msn.mgf | 3259 | 15032 | 19525 | 1780 | 39596 | 7860 | 14569 | 1098 |
| Summary | 170131 | 628675 | 524710 | 43123 | 1366639 | 409508 | 521589 | 56321 |

More library information are in Supplementary File 8. Spectype explanation: SINGLE_BEST_SCAN=highest TIC fragmentation scan is exported for each precursor in one injection, SAME_ENERGY=merging of spectra of the same precursor and collision energy in one injection, ALL_ENERGIES=merging of spectra of the same precursor and different collision energies in one injection (in our case mostly 3 energies) PSEUDO_MS2=whole $MS^n$ tree for one compound is merged into a pseudo-$MS^2$ scan.

# Reporting Summary

## Statistics

For all statistical analyses, confirm that the following items are present in the figure legend, table legend, main text, or Methods section.

| n/a | Confirmed | |
|---|---|---|
| ☒ | ☐ | The exact sample size (*n*) for each experimental group/condition, given as a discrete number and unit of measurement |
| ☒ | ☐ | A statement on whether measurements were taken from distinct samples or whether the same sample was measured repeatedly |
| ☒ | ☐ | The statistical test(s) used AND whether they are one- or two-sided <br> *Only common tests should be described solely by name; describe more complex techniques in the Methods section.* |
| ☒ | ☐ | A description of all covariates tested |
| ☒ | ☐ | A description of any assumptions or corrections, such as tests of normality and adjustment for multiple comparisons |
| ☒ | ☐ | A full description of the statistical parameters including central tendency (e.g. means) or other basic estimates (e.g. regression coefficient) AND variation (e.g. standard deviation) or associated estimates of uncertainty (e.g. confidence intervals) |
| ☒ | ☐ | For null hypothesis testing, the test statistic (e.g. *F*, *t*, *r*) with confidence intervals, effect sizes, degrees of freedom and *P* value noted <br> *Give P values as exact values whenever suitable.* |
| ☒ | ☐ | For Bayesian analysis, information on the choice of priors and Markov chain Monte Carlo settings |
| ☒ | ☐ | For hierarchical and complex designs, identification of the appropriate level for tests and full reporting of outcomes |
| ☒ | ☐ | Estimates of effect sizes (e.g. Cohen's *d*, Pearson's *r*), indicating how they were calculated |

*Our web collection on statistics for biologists contains articles on many of the points above.*

## Software and code

Policy information about availability of computer code

| Data collection | Xcalibur 4.5.445.18 |
|---|---|
| Data analysis | MZmine 4.0.8, https://github.com/mzmine/mzmine3, https://github.com/mzmine/mzmine_documentation, Python 3.10, https://github.com/corinnabrungs/msn_tree_library, https://github.com/corinnabrungs/msn_tree_library/tree/master/notebooks, https://gnps.ucsd.edu/ProteoSAFe/status.jsp?task=c05d34fb31ab4ee99293e722fb7eb83d |

For manuscripts utilizing custom algorithms or software that are central to the research but not yet described in published literature, software must be made available to editors and reviewers. We strongly encourage code deposition in a community repository (e.g. GitHub). See the Nature Portfolio guidelines for submitting code & software for further information.

## Data

Policy information about availability of data

All manuscripts must include a data availability statement. This statement should provide the following information, where applicable:
- Accession codes, unique identifiers, or web links for publicly available datasets
- A description of any restrictions on data availability
- For clinical datasets or third party data, please ensure that the statement adheres to our policy

All metadata files, one per compound library, and mzmine batch files for library processing were uploaded to the MERLIN (Mass spEctRal LIbrary Network) GitHub repository (https://github.com/merlin-ms) under the MIT license. All acquired flow injection–Orbitrap MSn files were deposited as .mzML and .raw files in the

Zenodo datasets: 10966280 (.mzML positive and negative), 10966404 (.raw positive), and 10967081 (.raw negative) under the CC BY 4.0 license, as well as to MassIVE MSV000094528 under the CCO 1.0 license. The mass spectral libraries included in MSnLib were deposited as .mgf and .json in the Zenodo dataset 11163380 under the CC BY 4.0 license. Here, each spectral library for the individual compound libraries is uploaded as MS2 only or the full MSn library. The MS2 libraries contain the best and merged spectra for all acquired MS2 spectra, including the pseudo-MS2, where the whole fragmentation tree is combined into a single spectrum. Polarities are kept separated, resulting in 4 entries for each compound library and library format. Additionally, the MS2 data of MCEBIO, MCESCAF, NIHNP, and OTAVPEP are uploaded as a reference library in GNPS (https://external.gnps2.org/gnpslibrary) in the form of a default gold-level library named MSNLIB-POSITIVE and MSNLIB-NEGATIVE. We recommend using the Zenodo libraries as they contain more metadata and link back to the original data by Universal Spectrum identifier (USI).
DrugBank lookup is an optional step, and the data is accessible on request. More information is available on the project website (https://go.drugbank.com).
DrugCentral lookup is an optional step, and their whole database can be downloaded as a PostgreSQL dump from the company's website (https://drugcentral.org).
LOTUS lookup is an optional step, and the whole LOTUS dataset is incorporated into WIKIDATA. The metadata cleanup script contains a prefect flow to download all relationships from WIKIDATA. Simply run the prepare_wikidata_lotus_data_prefect.py script (https://github.com/corinnabrungs/msn_tree_library).
Broad Institute lookup is an optional step. The drug information can be downloaded as a .txt file from the institute's website (https://repo-hub.broadinstitute.org/repurposing).
The FBMN result can be accessed on GNPS:
https://gnps.ucsd.edu/ProteoSAFe/status.jsp?task=c05d34fb31ab4ee99293e722fb7eb83d
Experimental public libraries were downloaded from the corresponding webpages:
GNPS = ALL_GNPS_NO_PROPAGATED (https://external.gnps2.org/gnpslibrary, access 08.12.2023)
MassBankEU = MASSBANK_NIST
(https://github.com/MassBank/MassBank-data/releases/tag/2023.11, access 08.12.2023)
MassBank North America = LC-MS/MS Spectra
(https://mona.fiehnlab.ucdavis.edu/downloads, access 08.12.2023)
The dataset used for the library evaluation can be found in MassIVE under MSV000096589. Here, we only used a subset of group 1 (group1_B*.mzML) for the analysis. The mzmine batch file is supplied as Supplementary File 9 and results of this evaluation as Supplementary File 10 and 11.

## Human research participants

Policy information about studies involving human research participants and Sex and Gender in Research.

| Reporting on sex and gender | N/A |
|---|---|
| Population characteristics | N/A |
| Recruitment | N/A |
| Ethics oversight | N/A |

Note that full information on the approval of the study protocol must also be provided in the manuscript.

# Field-specific reporting

Please select the one below that is the best fit for your research. If you are not sure, read the appropriate sections before making your selection.

☒ Life sciences    ☐ Behavioural & social sciences    ☐ Ecological, evolutionary & environmental sciences

For a reference copy of the document with all sections, see nature.com/documents/nr-reporting-summary-flat.pdf

# Life sciences study design

All studies must disclose on these points even when the disclosure is negative.

| Sample size | 9060 injections for 37,829 reference standards |
|---|---|
| Data exclusions | No |
| Replication | No |
| Randomization | No |
| Blinding | No |

# Reporting for specific materials, systems and methods

We require information from authors about some types of materials, experimental systems and methods used in many studies. Here, indicate whether each material, system or method listed is relevant to your study. If you are not sure if a list item applies to your research, read the appropriate section before selecting a response.

## Materials & experimental systems

| n/a | Involved in the study |
|-----|----------------------|
| ☒ | Antibodies |
| ☒ | Eukaryotic cell lines |
| ☒ | Palaeontology and archaeology |
| ☒ | Animals and other organisms |
| ☒ | Clinical data |
| ☒ | Dual use research of concern |

## Methods

| n/a | Involved in the study |
|-----|----------------------|
| ☒ | ChIP-seq |
| ☒ | Flow cytometry |
| ☒ | MRI-based neuroimaging |

