## [Peer Review File · Nature Methods]

MSⁿLib: Efficient generation of open multi-stage fragmentation mass spectral libraries

Corresponding Author: Dr Tomáš Pluskal

A version of this paper was originally rejected for publication by Nature Methods, however that decision was reconsidered after appeal by the authors.

Version 0:

Decision Letter:

28th Jun 2024

Dear Dr. Pluskal,

Your Resource entitled "Efficient generation of open multi-stage fragmentation mass spectral libraries" has now been seen by 2 reviewers, whose comments are attached. While they find your work of potential interest, they have raised a number of concerns which in our view are sufficiently important that they preclude publication of the work in Nature Methods, at least in its present form.

As you will see, the reviewers raise concerns about the current availability of the resource and how the data is presented within the resource. There are several technical concerns about the work. Reviewer #2 also mentions including stronger applications to showcase the value of the work. While we think it would be important to include examples to show how the data can be leveraged for new biological discovery, I'm happy to discuss the type of applications you might be able to add.

Should further data allow you to fully address these criticisms we would be willing to look at a revised manuscript (unless, of course, something similar has by then been accepted at Nature Methods or appeared elsewhere). This includes submission or publication of a portion of this work somewhere else. We hope you understand that until we have read the revised paper in its entirety we cannot promise that it will be sent back for peer-review.

If you are interested in revising this manuscript for submission to Nature Methods in the future, please contact me to discuss your appeal before making any revisions. Otherwise, we hope that you find the reviewers' comments helpful when preparing your paper for submission elsewhere.

Sincerely,
Arunima

Arunima Singh, Ph.D.
Senior Editor
Nature Methods

Reviewers' Comments:

Reviewer #1:

Remarks to the Author:

In this manuscript entitled "Efficient generation of open multi-stage fragmentation mass spectral libraries", the authors describe a new experimental and computational workflow (implemented in mzmime and in some additional Python scripts) to create a spectral library from approximately 20,600 small molecules, containing multi-stage fragmentation (MSⁿ, up to MS⁵) information as the key feature. The resulting resource (the library) will be made openly available (it is not the case at the time of the review), and then it will be possible to leverage it for compound annotation, and development of machine-learning models on

substructure-fragmentation patterns.

In the context of open data and metabolomics, this resource represents a key step forward. Although the methodology is sound and well explained (including some QC steps), however, the final product of this work (the actual library) is not made accessible by the authors at this point, and therefore, it could not be assessed directly. For instance, will retention time information be included there? How are the different ionisation polarities combined in the library? Which metadata is exactly included in the library? How traceable is the information back to the raw data? All pieces of information are really key, and they are not possible to assess directly at this point. The authors should also explain these details in the manuscript, or in the supplementary material.

It is also not fully clear the rationale behind the 5 compound libraries that were used as the basis. Could the authors elaborate more on this choice? And since the workflow seems to take only a few days to run, why did they stop with these ~20,600 compounds? In this context, there is key information missing about how it is planned that the spectral library will be maintained and sustained in the future. Will new compounds be added in a regular basis? Will the resource remain free to use? Will other instrument platforms be included? This information should be added to the manuscript.

Reviewer #2:

Remarks to the Author:

The authors have conducted a laudable project, making MSⁿ spectra publicly available. For too long, such data have been restricted to a specific private company. However, the publication suffers from a range of serious flaws. Most importantly, the authors only explain how data were generated, but they do not show the value of the data. The authors claim, without evidence, that the acquired spectra 'can be leveraged for compound annotation based on library matching, including substructures and training of machine learning models on substructure fragmentation patterns.' It is unclear if these statements are true, beyond the generic options that any MS/MS library would offer. First of all, then authors only used MSⁿ ion trees but did not seem to annotate each MSⁿ ion with a specific structure. Instead, MSⁿ ion trees seem to be restricted to elemental formulas, not refined to substructures. Without clear substructure annotations, how could then such data be used for machine-learning tools in substructure annotations? This schema is clearly inferior to the expert-driven annotation of substructures that the company HighChem performed, using their MassFrontier software that was built on peer-reviewed and published fragmentation pathways. Similarly, NIST has published tools for generating MS/MS libraries that annotate substructures, not just fragment ion formulas. Either tool would have been advantageous, instead of using the GNPS/ Sirius shortcuts as suggested by the authors. Moreover, it is not entirely clear to the reviewer how the MSⁿ spectra were stored? The authors state "Finally, a pseudo-MS2 spectrum was created by merging the whole MSⁿ tree." While such merged pseudo-MS2 spectrum has some debatable qualities and possible merits, no research was presented that would warrant the use of this merging strategy. The authors state that 177k MS2 spectra and 1.1m MSⁿ spectra were generated, but again, it is unclear how these 1.1m MSⁿ spectra were associated with true or likely structures and substructures? Next, the authors compared the spectra libraries to several open access libraries and two commercial libraries (mzCloud and NIST20). This is certainly a laudable step, however, the reviewer doubts that there are indeed 10k new compounds included in the new library that were missing before because (a) many more new entries are found in the NIST23 library that the authors failed to assess and (b) many other entries may be found in the commercial METLIN library that claims to include some 1m compounds. Last, the authors fail to showcase and compare the chemical space of the new library, either by itself or possibly by comparing against the chemical space offered by the other libraries. Here, the authors conduct a fallible and unusable MS-similarity network (a GNPS network blurb) instead of using genuine cheminformatics tools, starting with Tanimoto comparisons and (better) ending with graph-based approaches. MS/MS similarity of spectra is not identical to chemical similarity. In some cases, for specific compound classes and for highly related chemicals, MS/MS similarity may offer a surrogate for structure similarities (especially for spectra of unknown compounds, which was the main purpose for GNPS networks). But in general, MS/MS similarity networks cannot verify chemical similarities. The authors have used Tanimoto and MCES tools to qualify their own library entries, but do not show genuine structure overlaps (and compound class coverages) against all libraries (both open and commercial libraries, not just open libraries as stated in "80% of our spectral libraries remained new compared to the open libraries.")

As minor problem, the reviewer does not understand or agree with the strategy to remove low abundant ions "by removing signals below 2.5 times the lowest signal in each MS1-n scan". The authors claim this need to be due to an ion-trap based normalization. Neither this effect nor the 'Scan signal removal' feature in mzmine was explained in any detail, let alone validated or verified. Similarly, the authors stated to include a wide array of adducts at first, but then only to those ion (adducts) "with the highest detection frequencies". It is unclear to the reviewer which ion adducts were finally used in the library generation? Similarly, it is unclear to the reviewer why the authors used direct flow injections instead of cleaning up and separating isomers by (short) chromatography columns? The authors seem to assume that claims of purity of compounds (in chemical libraries) are true. The reviewer can assure the authors that this assumption is incorrect. There are a number of publications in the literature (which the authors do not acknowledge) that state otherwise.

** For Nature Portfolio general information and news for authors, see <http://npg.nature.com/authors>.

Version 1:

Decision Letter:

31st Jul 2024

Dear Dr. Pluskal,

Thank you for your letter asking us to reconsider our decision on your Resource, "Efficient generation of open multi-stage fragmentation mass spectral libraries". After careful consideration we have decided that we are willing to consider a revised version of your manuscript. However, there was not enough detail in the revision plan to be able to properly assess whether we would continue considering a revision, and so we will just take a look at the revised version when it is ready. We also encourage you to share the library with the reviewers so that they may assess it properly.

- * include a point-by-point response to our referees and to any editorial suggestions
- * please underline/highlight any additions to the text or areas with other significant changes to facilitate review of the revised manuscript
- * address the points listed described below to conform to our open science requirements
- * ensure it complies with our general format requirements as set out in our guide to authors at www.nature.com/naturemethods
- * resubmit all the necessary files electronically by using the link below to access your home page

Link Redacted

We hope to receive your revised paper within 4-6 weeks. If you cannot send it within this time, please let us know. In this event, we will still be happy to reconsider your paper at a later date so long as nothing similar has been accepted for publication at Nature Methods or published elsewhere.

OPEN SCIENCE REQUIREMENTS

REPORTING SUMMARY AND EDITORIAL POLICY CHECKLISTS

When revising your manuscript, please submit reporting summary and editorial policy checklists.

IMAGE INTEGRITY

DATA AVAILABILITY

CODE AVAILABILITY

Please include a "Code Availability" subsection in the Online Methods which details how your custom code is made available. Only in rare cases (where code is not central to the main conclusions of the paper) is the statement "available upon request" allowed (and reasons should be specified).

MATERIALS AVAILABILITY

SUPPLEMENTARY PROTOCOL

To help facilitate reproducibility and uptake of your method, we ask you to prepare a step-by-step Supplementary Protocol for the method described in this paper. We [encourage authors to share their step-by-step experimental protocols](https://www.nature.com/nature-research/editorial-policies/reporting-standards#protocols) on a protocol sharing platform of their choice and report the protocol DOI in the reference list. Nature Portfolio's protocols.io is a free-to-use and open resource for protocols; protocols deposited onto protocols.io are citable and can be linked from the published article. More details can found at [protocols.io](https://www.protocols.io/help/publish-articles).

ORCID

Sincerely,
Arunima

Arunima Singh, Ph.D.
Senior Editor
Nature Methods

Version 2:

Decision Letter:

28th Nov 2024

Dear Dr. Pluskal,

Your Resource, "Efficient generation of open multi-stage fragmentation mass spectral libraries", has now been seen by 2 reviewers. As you will see from their comments below, although the reviewers find your work of considerable potential interest, they have raised a number of concerns. We are interested in the possibility of publishing your paper in Nature Methods, but would like to consider your response to these concerns before we reach a final decision on publication.

We therefore invite you to revise your manuscript to address these concerns. In particular, we agree with the reviewer regarding the need for stronger applications/use cases for the resource. Alternatively, this work would also be suitable as a Brief Communication format, but we would still like to see some validation of the newly annotated metabolites.

Link Redacted

We hope to receive your revised paper within 4-5 weeks. If you cannot send it within this time, please let us know. In this event, we will still be happy to reconsider your paper at a later date so long as nothing similar has been accepted for publication at Nature Methods or published elsewhere.

OPEN SCIENCE REQUIREMENTS

REPORTING SUMMARY AND EDITORIAL POLICY CHECKLISTS

IMAGE INTEGRITY

DATA AVAILABILITY

All novel DNA and RNA sequencing data, protein sequences, genetic polymorphisms, linked genotype and phenotype data, gene expression data, macromolecular structures, and proteomics data must be deposited in a publicly accessible database, and accession codes and associated hyperlinks must be provided in the "Data Availability" section.

CODE AVAILABILITY

Please include a "Code Availability" subsection in the Online Methods which details how your custom code is made available. Only in rare cases (where code is not central to the main conclusions of the paper) is the statement "available upon request" allowed (and reasons should be specified).

For more information on our code sharing policy and requirements, please see: <https://www.nature.com/nature-research/editorial-policies/reporting-standards#availability-of-computer-code>

MATERIALS AVAILABILITY

SUPPLEMENTARY PROTOCOL

To help facilitate reproducibility and uptake of your method, we ask you to prepare a step-by-step Supplementary Protocol for the method described in this paper. We [encourage authors to share their step-by-step experimental protocols](https://www.nature.com/nature-research/editorial-policies/reporting-standards#protocols) on a protocol sharing platform of their choice and report the protocol DOI in the reference list. Nature Portfolio's protocols.io is a free-to-use and open resource for protocols; protocols deposited onto protocols.io are citable and can be linked from the published article. More details can found at [protocols.io](https://www.protocols.io/help/publish-articles).

ORCID

Sincerely,
Arunima

Arunima Singh, Ph.D.
Senior Editor
Nature Methods

Reviewers' Comments:

Reviewer #1 (Remarks to the Author):

In the revised version of the manuscript the authors have addressed my previous comments and clarified well the points that I raised. I have also checked the library files, and it is good that all the data is now publicly available.

There are two minor comments:

- The Zenodo dataset containing all the files should be documented in a better way. It would be good to add a README file or similar describing in some detail the content of each of the files.
- The USIs are included in the spectra files in Zenodo, but as far as I could see, they are not in the files uploaded into MassIVE.

It is also good that additional details about future plans have been added to the Discussion.

Reviewer #2 (Remarks to the Author):

The authors still do not provide actual use cases for this new repository. The authors argue that the repository itself is sufficiently new enough to justify its publication, which may or may not be the case. Analyzing samples that were matched against the new library, and detailing the correct identification of compounds that were not annotated before, should have been an easy task to convince readers of the possible benefit of the resource. Similarly, use cases how MSⁿ could be used to annotate substructures, with prospects for future developments on how such efforts could be automated by new computational tools (e.g. deep learning) would have been beneficial.

The authors still keep an unreadable and uninformative figure, Fig. 3 | Map of the chemical space covered by the MCEBIO MS2 library [...]. They now generated more informative Extended Data Fig. 3 | TMAP projections of the chemical space coverage for four mass spectral libraries compared to all chemical structures in those libraries. However, extended data Fig.3 fails to include the MCEBIO library. It would be advantageous to remove Fig and instead give a TMAP projection of the MCEBIO library against the other public libraries.

The authors asked the reviewer to add a reference that details how MS/MS libraries need to account for impure isomeric compounds (not necessarily only isobaric interferences). Such a reference was recently published as LibGen: Generating High Quality Spectral Libraries of Natural Products for EAD-, UVPD-, and HCD-High Resolution Mass Spectrometers in Analytical Chemistry. 2023 ;95(46):16810-8, but there are surely others. The authors should also include download dates for the different repositories, because they had not used NIST23 before, and it is unclear if other libraries (GNPS, MoNA etc) also had outdated download dates?

Version 3:

Decision Letter:

Our ref: NMETH-BC56340C

11th Mar 2025

Dear Dr. Pluskal,

Thank you for submitting your revised manuscript "Efficient generation of open multi-stage fragmentation mass spectral libraries" (N METH-BC56340C). It has now been seen by the original referees and their comments are below. While reviewer #2 still has some reservations about the impact of the applications shown in the paper, they find that the paper has improved in revision, and therefore we'll be happy in principle to publish it in Nature Methods, pending minor revisions to satisfy the referees' final requests and to comply with our editorial and formatting guidelines. However, the editorial team discussed the format for your paper and we think that instead of a Resource, a Brief Communication will be the more suitable option.

We are now performing detailed checks on your paper and will send you a checklist detailing our editorial and formatting requirements within two weeks or so. Please do not upload the final materials and make any revisions until you receive this additional information from us. The checklist will also include information/recommendations on how to change the format of your paper to a Brief Communication.

TRANSPARENT PEER REVIEW

ORCID

Sincerely,
Arunima

Arunima Singh, Ph.D.
Senior Editor
Nature Methods

Reviewer #2 (Remarks to the Author):

The authors have responded to several of the criticisms that were raised before. They have now added a section on a small use case of drugs metabolites by bacterial communities. The reviewer is not quite clear why this study (MSV000096589) was used, as one would have hoped the authors would use a study with natural compounds (e.g. foods, plants) instead of a drug-based study. The authors confirmed by their stringent criteria (6 matching ions, MS similarity >0.85) that these drugs were indeed added, and stated that existing libraries did not cover these drugs (or did not cover these drugs with those number of matching fragment ions).

While this use case is correctly identifying a few more compounds than before, matching drugs that were missed in classic libraries does not constitute the need for MS³ libraries. The authors do not make a clear case for the use of MS³ libraries, and the reviewer argues that these identifications would have likely been found if those drugs had been represented in classic MS/MS libraries.

Hence, the use case does not provide evidence for the necessity of using a MS³ method as such. Instead, the data provided by the authors only show the need to add more compounds to libraries (which is of course always true).

Less important, the authors argue that Figure 3 is necessary and valid. The presented arguments did not convince the reviewer.

Version 4:

Decision Letter:

9th Jul 2025

Dear Tomas,

I am pleased to inform you that your Brief Communication, "MSⁿLib: Efficient generation of open multi-stage fragmentation mass spectral libraries", has now been accepted for publication in Nature Methods. The received and accepted dates will be May 7, 2024 and July 9, 2025. This note is intended to let you know what to expect from us over the next month or so, and to let you know where to address any further questions.

Over the next few weeks, your paper will be copyedited to ensure that it conforms to Nature Methods style. Once your paper is typeset, you will receive an email with a link to choose the appropriate publishing options for your paper and our Author Services team will be in touch regarding any additional information that may be required.

Once proofs are generated, they will be sent to you electronically and you will be asked to send a corrected version within 48 hours. It is extremely important that you let us know now whether you will be difficult to contact over the next month. If this is the case, we ask that you send us the contact information (email, phone and fax) of someone who will be able to check the proofs and deal with any last-minute problems.

If, when you receive your proof, you cannot meet the deadline, please inform us at rjsproduction@springernature.com immediately.

If you are active on X or Bluesky, please e-mail me your and your coauthors' handles so that we may tag you when the paper is published.

To assist our authors in disseminating their research to the broader community, our SharedIt initiative provides you with a unique shareable link that will allow anyone (with or without a subscription) to read the published article. Recipients of the link with a subscription will also be able to download and print the PDF. As soon as your article is published, you will receive an automated email with your shareable link.

Please note that you and your coauthors may order reprints and single copies of the issue containing your article through Springer Nature Limited's reprint website, which is located at <http://www.nature.com/reprints/author-reprints.html>. If there are any questions about reprints please send an email to author-reprints@nature.com and someone will assist you.

Best regards,
Arunima

Arunima Singh, Ph.D.
Senior Editor
Nature Methods

** Visit the Springer Nature Editorial and Publishing website at www.springernature.com/editorial-and-publishing-jobs for more information about our career opportunities. If you have any questions please click here.**

Open Access This Peer Review File is licensed under a Creative Commons Attribution 4.0 International License, which permits use, sharing, adaptation, distribution and reproduction in any medium or format, as long as you give appropriate credit to the original author(s) and the source, provide a link to the Creative Commons license, and indicate if changes were made. In cases where reviewers are anonymous, credit should be given to 'Anonymous Referee' and the source.

Reviewers' Comments:

Reviewer #1:

Remarks to the Author:

In this manuscript entitled "Efficient generation of open multi-stage fragmentation mass spectral libraries", the authors describe a new experimental and computational workflow (implemented in *mzmine* and in some additional Python scripts) to create a spectral library from approximately 20,600 small molecules, containing multi-stage fragmentation (MS_n , up to MS_5) information as the key feature. The resulting resource (the library) will be made openly available (it is not the case at the time of the review), and then it will be possible to leverage it for compound annotation, and development of machine-learning models on substructure-fragmentation patterns.

In the context of open data and metabolomics, this resource represents a key step forward. Although the methodology is sound and well explained (including some QC steps), however, the final product of this work (the actual library) is not made accessible by the authors at this point, and therefore, it could not be assessed directly.

We thank the reviewer for acknowledging our effort that went into developing the MS^n method and automatic library generation workflow and creating this unique MS^n resource. Since the inception of this project, we have connected with many other researchers and were able to work together on sharing compounds, acquiring reference data, and helping others to build their own libraries or to implement our MS^n acquisition method in their labs.

Unfortunately, the Zenodo link was missing in our first submission. All the raw data, their converted *mzML* files, and MS^n libraries are uploaded both to Zenodo and to *MassIVE*/*GNPS*, two popular data repositories. In the meantime, we successfully added the subset of MS^2 spectra also to the *GNPS* spectral libraries. We apologize for this inconvenience and added the resources to the data availability section.

You will find all the data and libraries here:

MS^n libraries: [10.5281/zenodo.11163380](https://zenodo.org/record/11163380)

MS^2 Libraries (subset of MS^n Lib) on *GNPS* (*MSNLIB-POSITIVE* and *MSNLIB-NEGATIVE*):
<https://external.gnps2.org/gnpslibrary>

Data on *MassIVE*: <https://massive.ucsd.edu/ProteoSAFe/dataset.jsp?accession=MSV000094528>

Data Zenodo:

mzML: [10.5281/zenodo.10966280](https://zenodo.org/record/10966280)

raw positive: [10.5281/zenodo.10966404](https://zenodo.org/record/10966404)

raw negative: [10.5281/zenodo.10967081](https://zenodo.org/record/10967081)

The data availability section now reads:

“The mass spectral libraries included in MSⁿLib were deposited as .mgf and .json in the Zenodo dataset [11163380](https://doi.org/10.5281/zenodo.11163380) under the CC BY 4.0 license. Here, each spectral library for the individual compound libraries is uploaded as MS² only or the full MSⁿ library. The MS² libraries contain the best and merged spectra for all acquired MS² spectra, including the pseudo-MS², where the whole fragmentation tree is combined into a single spectrum. Polarities are kept separated, resulting in 4 entries for each compound library and library format. Additionally, the MS² data of MCEBIO, MCESCAF, NIHNP, and OTAVPEP are uploaded as a reference library in GNPS (<https://external.gnps2.org/gnpslibrary>) in the form of a default gold-level library named MSNLIB-POSITIVE and MSNLIB-NEGATIVE ”

For instance, will retention time information be included there?

This question about the available metadata is valid. We now added more information on the provided library, compound, and spectral metadata. Furthermore, the link to the library files in open-text formats is now provided.

Regarding the retention time, the library generation workflow itself was designed to extract m/z, retention time, ion mobility, collisional cross-section, and other descriptors, if available. This is useful when building LC-MS², GC-MS, or ion mobility-resolved libraries. MSⁿLib provides a time point for the MSⁿ acquisition, more for potential sanity checks that the scan was acquired on the flow injection transient. The whole library was acquired with a high-throughput flow injection method to allow for wide and deep MSⁿ tree acquisition with fast washout, therefore retention times are not available in MSⁿLib.

How are the different ionisation polarities combined in the library? Which metadata is exactly included in the library? How traceable is the information back to the raw data? All pieces of information are really key, and they are not possible to assess directly at this point. The authors should also explain these details in the manuscript, or in the supplementary material.

We appreciate the reviewer’s suggestion to include traceability in the manuscript. Backtracking library data is crucial for understanding their origin and the processing history. The lack of traceability in current spectral libraries was one of our key motivations for developing the automatic library generation workflow. This workflow tracks all parameters and processing steps and enables quick recreation of spectral libraries with adjusted settings. For MSⁿLib, we added the mzmine batch files used for processing which fully describe the spectral processing for reproducible library generation. This allows other researchers to use the same batch with modified settings if they need the library in a different format. To improve the traceability of spectra to raw data, each spectrum now contains a list of universal spectrum identifiers (USI; <https://www.nature.com/articles/s41592-021-01184-6>). USI is like a path into a public dataset containing a dataset ID, raw data file, and scan number. For merged spectra, this list contains all USI of all merged source scans. During the library generation, a dataset ID can be specified, otherwise a placeholder will be inserted that can be replaced later.

We have updated the main text to now describe the exported information:

“Positive and negative ionization modes were processed with separate mzmine batch configurations, due to the different adduct settings, resulting in individual libraries for each polarity. Each spectral library entry comprises metadata fields including the compound name, structural information, adduct type, collision energy, precursor m/z isolation window, quality check results, polarity, and spectrum type (single best scan or various merging types). Library-specific information can be added in the library export metadata, including the instrument type, contact details, and dataset ID, if data was uploaded to a public repository. The dataset ID also generates universal spectrum identifiers (USI) for each spectral library entry. USI facilitates traceability of FAIR data similar to a file path into the public domain, denoting the dataset, filename, and scan number. Merged scans contain all USI of their source scans. The available metadata is useful to filter the MSnLib into subsets, e.g., for specific instruments.”

It is also not fully clear the rationale behind the 5 compound libraries that were used as the basis. Could the authors elaborate more on this choice? And since the workflow seems to take only a few days to run, why did they stop with these ~20,600 compounds? In this context, there is key information missing about how it is planned that the spectral library will be maintained and sustained in the future. Will new compounds be added in a regular basis? Will the resource remain free to use?

We agree with the reviewer that the rationale for the used compound libraries should be explained in more detail, which also highlights one of the bottlenecks for mass spectral library generation. For our MSⁿLib, we used all compound libraries that were available to us. In the future, we want to expand this resource by adding more compound libraries. Unfortunately, thousands of standards are quite expensive to purchase and can cost tens to hundreds of thousands of euros. This is why we added the section below in the discussion to highlight the need for collaborations to share compound libraries and link synthesis labs and mass spectrometry labs, to expand outside of the mass spectrometry community. Furthermore, we want to encourage the metabolomics community to share data openly. This resource will remain free of charge and fully open. We are actively expanding the collection of compounds and MSⁿ trees and continue uploading them to multiple repositories (zenodo, MassIVE/GNPS). The library is now nearly doubled in size with 37,829 compounds and 34,413 unique structures acquired.

“Looking ahead, we aim to acquire additional compound libraries and expand the covered chemical space by continuously adding to the free and open MSⁿLib resource. The primary challenge in constructing large-scale mass spectral libraries is access to compound libraries, which are often costly and required in small quantities. This necessitates collaboration. We are grateful to our collaborators, who have already provided 7 compound libraries, and are actively reaching out to community members to share commercial, purified, or synthesized compounds.”

Will other instrument platforms be included? This information should be added to the manuscript.

For the MSⁿLib we relied on the MSⁿ capabilities of our Orbitrap ID-X Instrument. There are not many other instruments with MSⁿ modes. In a related library building project, the Mass spEctral Library Network (MERLIN), we are building a collaborative network for sharing compounds, acquisition time, data, and software to broaden the scope of open MS libraries. In this network, we already have plenty of labs contributing with other instruments, fragmentation techniques, and software. Others are synthesizing or sending compounds to be measured. This project also includes retention time, retention index, ion mobility, CCS, and other sample introduction systems like MALDI spot analysis. All data will be traceable, open, and free.

We changed the discussion to describe why the library was built with a single instrument and what our plans are to widen the scope to other libraries:

“MSⁿLib data has been acquired on a single Orbitrap ID-X instrument because of its MSⁿ capabilities, however, we plan to build other MS² libraries by including data from various instruments and fragmentation techniques, also capturing additional identifiers like retention time and ion mobility. This will enhance the diversity and utility of open mass spectral libraries.”

Reviewer #2:

Remarks to the Author:

The authors have conducted a laudable project, making MSⁿ spectra publicly available. For too long, such data have been restricted to a specific private company. However, the publication suffers from a range of serious flaws. Most importantly, the authors only explain how data were generated, but they do not show the value of the data. The authors claim, without evidence, that the acquired spectra 'can be leveraged for compound annotation based on library matching, including substructures and training of machine learning models on substructure fragmentation patterns.' It is unclear if these statements are true, beyond the generic options that any MS/MS library would offer.

We thank the reviewer for their thorough evaluation of our manuscript and for highlighting the uniqueness of this open MSⁿ library. Indeed, this Resource article focuses more on the resource itself. This includes the development and optimization of the data acquisition and the data processing workflows. The key aim of this paper is to showcase the library building workflow, from metadata curation over data acquisition to automatic library generation. To our best knowledge, there is no other automatic workflow capable of exporting MSⁿ tree libraries. More generally, software to handle MSⁿ data is rather limited, which might also be due to the limited availability of MSⁿ-ready instruments. We think that providing this computational workflow and resource will enable the development and testing of new MSⁿ software.

Regarding the value of the data, MSⁿLib is already one of the largest free and open mass spectral libraries and adds 21,000 new molecular structures compared to other open libraries. Furthermore, MSⁿLib provides a unique insight into the fragmentation of structures, by showing which fragment may (not) produce another fragment. So the utilities go far beyond a simple MS² library. While we are still exploring many of these implications and opportunities in our machine-learning applications, we want to make the data available as soon as possible and describe the whole workflow.

Furthermore, we are also collaborating with various researchers to leverage the data in different applications, and we added those resources to the text. From this research, one publication was released one month ago, dealing with the library cleaning of publicly available spectra, including GNPS, MassBankEU, and MoNA, together with our MSⁿLib (<https://doi.org/10.1186/s13321-024-00878-1>). The results show that all public libraries suffer from wrongly annotated spectra or wrong metadata, that the authors either cleaned or even removed. In this comparison, almost all spectra from MSⁿLib were retained, with few structures not parsable by rdkit or uncommon adduct types not handled in their cleaning pipeline. Another preprint by our group, describing a machine learning model that was trained on unannotated MS² spectra was evaluated with MSⁿLib for predicting fluorinated molecules and other prediction tasks (<https://chemrxiv.org/engage/chemrxiv/article-details/6626775021291e5d1d61967f>). Additionally, we are also collaborating with the authors of MS2DeepScore to retrain their models with the new spectral libraries. We are trying to estimate how the performance changes by adding the pseudo MS² spectra which are merged across all MSⁿ into the training data.

We added those publications to the manuscript's discussion. First, about the external evaluation of the quality of MSⁿLib:

“ All necessary information for regenerating the MSⁿ library, namely the data, compound metadata, and mzmine workflow configuration, is openly shared. This is a paradigm shift in the generation of spectral libraries, leading to greater traceability of the library entries to raw data using the Universal Spectrum Identifier (USI) with transparent processing parameters. A recent study validated our library quality by checking compound and spectral metadata in all public spectral libraries. These checks removed over 31,000 spectra each from GNPS, MoNA and MassBankEU, while MSⁿLib failed checks for 58 entries.³⁵ This shows that automatically curated libraries may improve the overall quality compared to historically grown libraries that relied on manual curation by contributors and often lost links to the processing history and original raw data.”

Then we describe its prospect for machine learning and linking out to other projects that are already using MSⁿLib.

“We expect that the demonstrated high-throughput method will facilitate rapid growth in the public availability of reference spectra and that this unique resource will stimulate the development of new machine learning-based tools leveraging the spectral tree dependency of fragment ions across multi-stage fragmentation. Newer models for MS2DeepScore¹⁸ and MS2Query¹⁹ are already trained with the new MSⁿLib (MS² only) together with GNPS data. The DreaMS project uses MSⁿLib for evaluation and fine-tuning of their model for specialized tasks, such as predicting fluorinated compounds.³⁴”

First of all, then authors only used MSⁿ ion trees but did not seem to annotate each MSⁿ ion with a specific structure. Instead, MSⁿ ion trees seem to be restricted to elemental formulas, not refined to substructures. Without clear substructure annotations, how could then such data be used for machine-learning tools in substructure annotations? This schema is clearly inferior to the expert-driven annotation of substructures that the company HighChem performed, using their MassFrontier software that was built on peer-reviewed and published fragmentation pathways. Similarly, NIST has published tools for generating MS/MS libraries that annotate substructures, not just fragment ion formulas. Either tool would have been advantageous, instead of using the GNPS/Sirius shortcuts as suggested by the authors.

This is a very important comment, as substructure annotation would be a final goal. To our experience and knowledge, there is no tool that can predict substructures for MS fragmentation experiments with high certainty. Most tools fail for more complex rearrangements or just provide a long list of fragment candidates for each signal. A recent publication also concluded that most substructures were incorrect (<https://www.nature.com/articles/s42004-024-01112-7>). MassFrontier is a great resource but still error-prone and provides multiple candidates for each signal and there is no clear way to make a decision for one substructure. We used MassFrontier to predict potential fragment structures for the first 20,000 compounds, and we may use this information for further filtering or annotation in the future - but at the moment, we are uncertain of the quality of the substructure annotation. For this reason, we are not including them in the current resource.

The work done by HighChem and the mzCloud team is a huge effort that is consuming incomparable amounts of money for the manual curation of MSⁿ trees. They are also offering automatically created libraries without manual curation and without substructure annotations, which still depend on manual curation. Together with the computational MS community, we can add substructures in the future. However, this is too much for a small team. Now that the resource is openly available, we already received great new ideas to enhance MSⁿLib and our library building workflow.

We did check some MSⁿ trees manually with MassFrontier, but this step is very time-consuming and still error-prone. Together with the mzmine developers, we are working on a method to include the substructure annotation within mzmine spectral library matching and library generation. Users could then generate substructures with their preferred tool, like MassFrontier, and use the substructures as a filter for how many signals they can explain. But this task needs more time for validation. Once this is included and library annotations can be checked not only by subformula but substructure, library data can be regenerated and updated. This shows the need for living library data that can be dynamically rebuilt once we have more knowledge and more quality checks included in our automatic workflow.

Even without substructure-level annotations, MSⁿ trees still add valuable information. MS² links fragment ions to one parent m/z and MSⁿ creates more links within fragment signals, showing which fragment may or may not produce another fragment ion. While this is already used by MS experts in structure elucidation, machine learning models can use this information to retrain the relationship between fragment signals. This way, embedding models like DreaMS and MS2Deepscore may improve their spectral clustering and internal representation of mass spectra as vectors. The DreaMS model, for instance, was trained on unannotated data and still gained an internal representation of mass spectra that mapped chemical relationships ([10.26434/chemrxiv-2023-kss3r-v2](https://doi.org/10.26434/chemrxiv-2023-kss3r-v2)).

We added an explanation pointing to the extensive manual validation needed for substructures to the discussion:

“Substructure prediction by MassFrontier, CFM-ID, or other tools may add additional value to the MSⁿLib resource. However, substructure prediction requires extensive manual validation and we invite the mass spectrometry community to join our effort in developing solutions.”

Moreover, it is not entirely clear to the reviewer how the MSⁿ spectra were stored? The authors state “Finally, a pseudo-MS² spectrum was created by merging the whole MSⁿ tree.” While such

merged pseudo-MS2 spectrum has some debatable qualities and possible merits, no research was presented that would warrant the use of this merging strategy. The authors state that 177k MS2 spectra and 1.1m MSⁿ spectra were generated, but again, it is unclear how these 1.1m MSⁿ spectra were associated with true or likely structures and substructures?

Unfortunately, the Zenodo link was missing in our first submission. All the raw data, their converted mzML files, and MSⁿ libraries are uploaded both to Zenodo and to MassIVE/GNPS, two popular data repositories. In the meantime, we successfully added the subset of MS² spectra also to the GNPS spectral libraries. We apologize for this inconvenience and added the resources to the data availability section.

You will find all the data and libraries here:

MSⁿ libraries: [10.5281/zenodo.11163380](https://zenodo.org/record/11163380)

MS² Libraries (subset of MSⁿLib) on GNPS (MSNLIB-POSITIVE and MSNLIB-NEGATIVE): <https://external.gnps2.org/gnpslibrary>

Data on MassIVE: <https://massive.ucsd.edu/ProteoSAFe/dataset.jsp?accession=MSV000094528>

Data Zenodo:

mzML: [10.5281/zenodo.10966280](https://zenodo.org/record/10966280)

raw positive: [10.5281/zenodo.10966404](https://zenodo.org/record/10966404)

raw negative: [10.5281/zenodo.10967081](https://zenodo.org/record/10967081)

The data availability section now reads:

“The mass spectral libraries included in MSⁿLib were deposited as .mgf and .json in the Zenodo dataset [11163380](https://zenodo.org/record/11163380) under the CC BY 4.0 license. Here, each spectral library for the individual compound libraries is uploaded as MS² only or the full MSⁿ library. The MS² libraries contain the best and merged spectra for all acquired MS² spectra, including the pseudo-MS2, where the whole fragmentation tree is combined into a single spectrum. Polarities are kept separated, resulting in 4 entries for each compound library and library format. Additionally, the MS2 data of MCEBIO, MCESCAF, NIHNP, and OTAVPEP are uploaded as a reference library in GNPS (<https://external.gnps2.org/gnpslibrary>) in the form of a default gold-level library named MSNLIB-POSITIVE and MSNLIB-NEGATIVE”

In mzmine, MSⁿ trees can be visualized with the tree visualizer. Most software is developed to handle MS² data and is incapable of handling MSⁿ trees. This is most likely also due to lacking public resources like MSⁿLib but also due to the rarity of MSⁿ-ready instruments. We hope this will change in the future, as multi-stage fragmentation can provide more insights, boosting manual structure elucidation with great prospects for in silico applications. This is why we include the pseudo MS² spectra that merge all MSⁿ information to be used in MS²-only tools, like SIRIUS and GNPS. We already compared different fragmentation techniques against each other, including CID, HCD, UVPD, and EAD (on a qTOF-MS instrument). For instance, comparing the merged CID or HCD spectra sometimes provided higher matching scores against UVPD, enabling greater confidence in the annotation. With MSⁿ we push the fragmentation to smaller fragments, which may be more similar to other fragmentation techniques. The spectype metadata field in library entries allows for easy filtering of MSⁿLib to exclude the merged spectra or the pseudo MS² spectra for specific applications. We added exported metadata and additional information in the main and online methods sections:

“Positive and negative ionization modes were processed with separate mzmine batch configurations, due to the different adduct settings, resulting in individual libraries for each polarity. Each spectral library entry comprises metadata fields including the compound name, structural information, adduct type, collision energy, precursor m/z isolation window, quality check results, polarity, and spectrum type (single best scan or various merging types). Library-specific information can be added in the library export metadata, including the instrument type, contact details, and dataset ID, if data was uploaded to a public repository. The dataset ID also generates universal spectrum identifiers (USI) for each spectral library entry. USI facilitates traceability of FAIR data similar to a file path into the public domain, denoting the dataset, filename, and scan number. Merged scans contain all USI of their source scans. The available metadata is useful to filter the MSⁿLib into subsets, e.g., for specific instruments.”

“Spectral library export to .json, .mgf, or .msp formats

1. Scoring of the precursor isolation purity-% of MSⁿ spectra based on the preceding and following MS¹ scan. Chimeric spectra are flagged in the output file.
 - a. Exporting the best spectrum for each precursor and energy (highest TIC), no SPECTYPE information or “SINGLE_BEST_SCAN” in library file
2. Merging of spectra, SPECTYPE information in the library file:
 - a. “SAME_ENERGY”: Each individual fragmentation energy, when triggered multiple times in the same sample
 - b. “ALL_ENERGIES”: All fragmentation energies for individual precursors (3 energies in our method, using the merged same_energy if available otherwise the best one)
 - c. “ALL_MSⁿ_TO_PSEUDO_MS²”: Combining the full MSⁿ tree of a compound ion into a pseudo-MS² spectrum
3. Filtering of spectra based on a minimum of two signals above the noise threshold”

Next, the authors compared the spectra libraries to several open access libraries and two commercial libraries (mzCloud and NIST20). This is certainly a laudable step, however, the reviewer doubts that there are indeed 10k new compounds included in the new library that were missing before because (a) many more new entries are found in the NIST23 library that the authors failed to assess and (b) many other entries may be found in the commercial METLIN library that claims to include some 1m compounds.

We thank the reviewer for highlighting the structural comparison step and that we should include NIST23 as there was a significant increase in the total number of compounds. NIST was very supportive and shared the structure table with us. However, this was not the case for METLIN. We reached out to them several times, without any answer. We also know from other collaborators that there is no information available which compounds are included in the METLIN database. Even after buying access to METLIN, customers do not see a list of compounds.

When comparing the unique compounds with all other main resources, we were also surprised by the number, which is now even higher (21k) after adding new compounds to MSⁿLib. Before comparing the first InChIKey block, we cleaned the structures of all resources using the same modified script from ChEMBL (structural cleaning, standardization, and InChIKey calculation). The first block of the InChIKey disregards stereochemistry. We decided that this is the best way to compare the chemical space, considering that mass spectrometry most often does not differentiate between stereoisomers.

We added a sentence in the main text to highlight the structure cleaning, standardization, and comparison:

“Prior to the comparison, we cleaned and standardized the structure of all resources in the same way and calculated the InChIKey from here. The comparison is based on the first InChIKey block, removing stereochemistry.”

Last, the authors fail to showcase and compare the chemical space of the new library, either by itself or possibly by comparing against the chemical space offered by the other libraries. Here, the authors conduct a fallible and unusable MS-similarity network (a GNPS network blurb) instead of using genuine cheminformatics tools, starting with Tanimoto comparisons and (better) ending with graph-based approaches.

We want to thank the reviewer for their advice to add a chemical space projection. We addressed this in our revised manuscript by adding a TMAP projection for the chemical space coverage. In our experience, the employed fingerprint and TMAP projection provide a good representation of the chemical space. The structures for NIST23 and mzCloud are not publicly available - therefore we cannot share those structure lists along with the figures. The TMAP plots demonstrate how MSnLib and the other libraries are complementary and contain different compound classes.

The chemical space coverage was missing in the manuscript and is now included as Extended Data Fig. 3. and the main text.

Extended Data Fig. 3 | TMAP³² projections of the chemical space coverage for four mass spectral libraries compared to all chemical structures in those libraries. The libraries are divided in a MSⁿLib, b Open libraries, including GNPS, MoNA, and MassBankEU, c NIST 23 and d mzCloud. Each node represents an individual structure based on canonical SMILES. Compounds included in the corresponding library are highlighted in magenta and drawn on top of compounds from other libraries in grey. Same structure can be included in multiple libraries.

“The chemical space coverage is illustrated in Extended Data Fig. 3 as a TMAP projection.³² This projection includes all chemical structures from MSⁿLib, the three open spectral libraries combined, mzCloud, and NIST23, representing the entire chemical space covered by spectral libraries. Each spectral library encompasses a distinct area, with NIST23 demonstrating the most extensive coverage due to its high number of unique compounds. The chemical space TMAP projections reflect the uniqueness of each spectral library and highlight in grey branches how complementary their coverage is. By integrating the newly created MSⁿLib with the three largest open spectral libraries, we achieve a comparable chemical space coverage.”

MS/MS similarity of spectra is not identical to chemical similarity. In some cases, for specific compound classes and for highly related chemicals, MS/MS similarity may offer a surrogate for structure similarities (especially for spectra of unknown compounds, which was the main purpose for GNPS networks). But in general, MS/MS similarity networks cannot verify chemical similarities.

We agree that MS/MS similarity is quite limited in its scope but it can be useful as a surrogate for structural similarity for transformation products. We used molecular networking and spectral library matching against all public spectral libraries as a sanity check for our automatic library building workflow. First, the spectral similarity (Fig. 3) demonstrates that there are enough fragment signals (>=4) in our MSⁿLib entries to establish networking edges. The spectral quality is also highlighted by Extended Data Fig. 1 and 2.

The networks were used in combination with spectral library matches against other libraries using the MCES and Tanimoto similarity to compare the actual structure with the top matches. In the resulting network (Fig. 3 and Supplementary File 7), similar spectra cluster together and show their structurally closest hit in other libraries by color coding. This highlights the number of new spectra over what was available previously.

The authors have used Tanimoto and MCES tools to qualify their own library entries, but do not show genuine structure overlaps (and compound class coverages) against all libraries (both open and commercial libraries, not just open libraries as stated in “80% of our spectral libraries remained new compared to the open libraries.”)

This is a good point and more clarification in the text is needed. The Tanimoto similarity and MCES were used to validate the matches against public libraries. This can be seen in two ways, first, it is used as a sanity check if the automatic library building workflow results in exporting correct spectra that are similar to already existing reference data. If similar or the same structures are matched, the library spectra are meaningful. Second, spectra that remain unmatched point to spectra with completely new information, although other spectra of the corresponding compound may be available. For a comparison based on structures, we used the UpSet plot-based direct matches on the first block of the InChIKey, ignoring stereochemistry. Further, we added the TMAP projection to better visualize the chemical space.

We added this sentence in the main text for clarification:

"It is important to note that the same compound may be in the public database, but differences in instruments, fragmentation energies, or settings can affect the library match."

The library matching was only applied to open libraries, as those data can be downloaded and the same workflow and parameters can be set. We also added this in the main text:

"Here, we limited the matching to open libraries, enabling the same spectral matching parameters and workflow."

As minor problem, the reviewer does not understand or agree with the strategy to remove low abundant ions "by removing signals below 2.5 times the lowest signal in each MS1-n scan". The authors claim this need to be due to an ion-trap based normalization. Neither this effect nor the 'Scan signal removal' feature in mzmine was explained in any detail, let alone validated or verified.

We thank the reviewer for the critical review and comment. We added clarification to our text now pointing to the Extended Data Fig. 2a, which illustrates that noise intensity fluctuations are correlated with 1/injection time. The Orbitrap aims to measure the same total number of ions in every scan and normalization divides the intensity by the injection time. Other processing strategies use an absolute noise level, which is not suitable for Orbitrap-based systems. Therefore, a simple factor of the lowest signal noise removal was used, as it is implemented in mzmine. The effect of this filter is shown in Supplementary Fig. 3, where the lowest signal is highlighted (red), together with the factor of 2.5 (yellow). Users who require MSⁿLib in a different flavor can reprocess the library with the provided data, metadata, and modified configuration files to produce a library that contains all noise signals. It is as easy as removing the noise level.

“[...] all spectra were denoised by removing signals below 2.5 times the lowest signal in each MS¹-n scan (see **Supplementary Fig. 3**), accounting for the varying noise level due to ion trap-based normalization. This is illustrated in the **Extended Data Fig. 2a**, where the lowest signal intensity increases with shorter filling times due to the normalization processing of the vendor. This fact prevents the use of an absolute noise level.”

The scan signal removal filter should be only used in case of reoccurring static noise or contamination signals to clean the spectra. We only used it in the case of abundant static noise signals that are already flagged by the vendor (Thermo Fisher). These signals are also removed in other tools like FreeStyle, if not explicitly selected to show. Even the ThermoRawFileParser removes these signals from MS¹ and MS² scans during the conversion to .mzML. However, they fail to remove them from MS³⁻ⁿ scans. Accordingly, we developed a new module in mzmine to remove static noise, which is instrument or even lab-specific. The effect is discussed in Supplementary Fig. 4, and we added this information to the main text:

“In less intense MSⁿ scans, broadly distributed noise signals were observed with an irregularly high mass defect for low m/z values (see Supplementary Fig. 4). The instrument vendor typically flags such artifacts in the spectra, but the ThermoRawFileParser failed to remove those artifacts in MS³⁻ⁿ during data conversion. Therefore, mzmine’s ‘Scan signal removal’ was applied to remove reoccurring known contaminant signals. Due to their uncommon mass defects, real spectral data should remain unaffected. This step should be used with caution, and only abundant frequently detected signals should be removed that cannot be explained by a subformula from the original structures.”

Similarly, the authors stated to include a wide array of adducts at first, but then only to those ion (adducts) “with the highest detection frequencies”. It is unclear to the reviewer which ion adducts were finally used in the library generation?

We thank the reviewer for pointing out the missing information. This was now added to the main text:

“For MSⁿLib, we included six adduct types in positive ion mode as [M], [M]⁺, [M+H]⁺, [M-H₂O]⁺, [M-H₂O+H]⁺, [M-2H₂O+H]⁺, [M+NH₄]⁺, and [M+Na]⁺ and five adducts in negative ion mode, namely [M], [M]⁻, [M-H]⁻, [M+Cl]⁻, and [M+FA]⁻. In mzmine, [M] refers to an ion that is intrinsically charged and might match compounds like quaternary amines.”

Similarly, it is unclear to the reviewer why the authors used direct flow injections instead of cleaning up and separating isomers by (short) chromatography columns? The authors seem to assume that claims of purity of compounds (in chemical libraries) are true. The reviewer can assure the authors that this assumption is incorrect. There are a number of publications in the literature (which the authors do not acknowledge) that state otherwise.

We thank the reviewer for their comment and are interested in recommendations of publications discussing compound purity. We agree fully that the chemical standards and isolated natural products in MSⁿLib may not be pure by any means. We even mixed up to 10 compounds to cut down on the data acquisition time, consequently allowing for the high throughput acquisition of libraries. However, we do not expect many avoidable interferences with the same m/z in our mixes. To identify interferences with similar but not exact m/z, we included the precursor purity score that checks the main precursor signal intensity in MS¹ divided by a sum of all signals in the isolation window. However, from the data alone, it will be hard to estimate or detect interferences with the same molecular formula. Even liquid chromatography will not solve the issue of stereoisomers and other interferences, which may require extensive method development and various column types. We agree that liquid chromatography is valuable for the retention time information, and we are collaborating with other groups to create retention time databases and other spectral libraries acquired by LC-MS².

The challenge with MSⁿ data acquisition lies in the extended time required to obtain a full fragmentation tree, which is often performed using flow injection or direct injection analysis. As highlighted in Figure 1b, the maximum time for a full tree with our method was 13 seconds, although most ions required less time due to MSⁿ trigger constraints. Our flow injection method offered unique high throughput with 50% of the time spent with actual data and fast wash-out times to avoid carryover. Fast chromatography delivers peaks that are too narrow for a full MSⁿ tree acquisition for our library.

For another project, we acquired the MCEBIO and NIHNP compound libraries also by RP-LC-MS² with a short 8.7 min gradient. A total of 1581 injections in 10 days runtime just for positive ion mode and one column type and condition - without the possibility for deep MSⁿ. The same 1581 injections took less than 4 days by flow injection-MSⁿ. We see these acquisition methods and the resulting data as complementary and want to acquire both in the future. The data processing workflow can handle any kind of data.

We added the following statement to the main text:

“Finally, we emphasize that the libraries were acquired without chromatographic separation by flow injection-MSⁿ, essential for obtaining deep and wide MSⁿ trees in a high throughput workflow. The compound libraries, sourced from various companies, were not individually tested for quality. Consequently, structural isomers and other contaminants may be present during acquisition, annotation, and in the final MSⁿLib.”

Reviewer #1 (Remarks to the Author):

In the revised version of the manuscript the authors have addressed my previous comments and clarified well the points that I raised. I have also checked the library files, and it is good that all the data is now publicly available.

We appreciate the positive assessment of the revised manuscript by the reviewer.

There are two minor comments:

- The Zenodo dataset containing all the files should be documented in a better way. It would be good to add a README file or similar describing in some detail the content of each of the files.

We provided a detailed description for each Zenodo record that is visible at the record webpage. We had to separate the raw data, converted data, and spectral libraries into individual Zenodo records because of Zenodo record size constraints. The spectral library files are available in two common library formats (.mgf and .json) and named accordingly, taking into account the seven compound libraries, the two ion modes, and the separation into MS²-only and MSⁿ files. We created a separate MS²-only library because this is a common use case for tools that do not support MSⁿ trees.

Zenodo offers great APIs to access the description and further details of the records in json format. One example is:

<https://zenodo.org/api/records/11163380> (redirects to the latest version of the record ID)

Otherwise, the Zenodo website renders this information in a human-readable form:

<https://zenodo.org/records/11163380>

- The USIs are included in the spectra files in Zenodo, but as far as I could see, they are not in the files uploaded into MassIVE.

We thank the reviewer for checking the uploaded files so thoroughly. We are not 100% certain what specific files this comment refers to, but here we provide an overview of the upload status:

MassIVE: During the manuscript revision, there was an issue with the data upload to MassIVE, and we resolved the issue in collaboration with the MassIVE team. Now, all final raw data and mzML files have been uploaded to this repository under accession ID MSV000094528 (as an alternative to Zenodo). The MassIVE repository itself does not contain any spectral libraries.

GNPS library: We uploaded all the MS2 spectra of MSnLib to GNPS via batch library submission. GNPS has a strict format for the fields and values allowed as metadata. Currently, there is no way to provide a USI link to the raw data for library spectra on GNPS.

This library part is accessible online:

<https://external.gnps2.org/gnpslibrary> named MSNLIB-POSITIVE and MSNLIB-NEGATIVE.

Anyone can browse the libraries online or download them:

<https://gnps.ucsd.edu/ProteoSAFe/gnpslibrary.jsp?library=MSNLIB-POSITIVE>

GNPS automatically generates a library ID and corresponding USI that point to these library entries:

<https://gnps.ucsd.edu/ProteoSAFe/gnpslibraryspectrum.jsp?SpectrumID=CCMSLIB00012274321#%7B%7D>

Overall, we recommend using the Zenodo libraries as they contain much more metadata and link back to the original raw data. We added this information to the Data availability section as a recommendation.

It is also good that additional details about future plans have been added to the Discussion.

We thank the reviewer for the great and constructive review. It helped us improve the manuscript.

Reviewer #2 (Remarks to the Author):

The authors still do not provide actual use cases for this new repository. The authors argue that the repository itself is sufficiently new enough to justify its publication, which may or may not be the case. Analyzing samples that were matched against the new library, and detailing the correct identification of compounds that were not annotated before, should have been an easy task to convince readers of the possible benefit of the resource.

We have carefully considered the reviewer's comments and have addressed the missing use case in this revision. We searched the Massive repository for data acquired by a different lab as an actual use case of how researchers will use the MSⁿLib library later in their studies. For spectra matching, we used strict criteria, e.g., matching of 6 signals and cosine similarity of 0.85, to detect only high-quality spectral matches. We compared our library to the three combined open spectral libraries (GNPS, MoNA, MassBankEU). First, we evaluated MSnLib and the open libraries individually, resulting in 80 and 129 matches, respectively, for a dataset with 6913 features with MS2 data. If we combine the annotations of both libraries, the number of annotations increases to 150, resulting in 21 additional matches with the new MSⁿLib.

We added this analysis to the main text:

“The newly generated MSⁿLib was further evaluated by matching against a public metabolomics dataset (MSV000096589) in combination with the other open spectral libraries. This dataset analyzes bacterial community cultures incubated with various drugs. After mzmine data processing, the feature table contained 9078 features with 6913 including MS² experiments. The library matching parameters were set for higher confidence with at least 6 matching signals and a weighted cosine similarity of ≥ 0.85 to only include high-quality spectral matches. The other open libraries (MoNA, GNPS, MassBank EU) and MSnLib yielded 129 and 80 annotated features, respectively. The combined total number of annotations was 150 with 21 unique annotations by MSⁿLib. The unique annotations were evaluated and compared against the list of 142 added drugs. Based on the first InChIKey block, 14 annotations correspond directly to the added drugs. The seven remaining matches may originate from microbial metabolism, e.g., ufiprazole as a metabolite of omeprazole, and were not further investigated. Library matching results and spectral mirror plots for these matches are collected in Supplementary File 10 and Supplementary File 11. These results on a real dataset further validate the quality of MSⁿLib entries and the complementary nature to other open libraries.”

Similarly, use cases how MSⁿ could be used to annotate substructures, with prospects for future developments on how such efforts could be automated by new computational tools (e.g. deep learning) would have been beneficial.

We thank the reviewer for this comment. We are citing a selection of articles that successfully use MSⁿ data for structure elucidation. Multi-stage fragmentation is a very established method for this due to the additional fragmentation information and tree-like fragmentation routes. Below, we highlight a selection of articles, but there are many more, focused on different compound classes.

1. <https://link-springer-com.uaccess.univie.ac.at/article/10.1007/s11306-011-0363-7>
2. <https://analyticalsciencejournals.onlinelibrary.wiley.com/doi/10.1002/rcm.6340>
3. https://www.sciencedirect.com/science/article/pii/S0165993615000916?casa_token=mFSRBNML7WYAAAAA:3YhEckd9POtCEit7cZewZPfYtDQTGrxamo4ceCwc311DhBA1r1q-pjl65wfPrHyWxe8PebQkEhM
4. <https://www.sciencedirect.com/science/article/pii/S2667145X23000329>
5. https://pubs.acs.org/doi/full/10.1021/ac2034216?casa_token=iklhH6LDUcwAAAAA%3A16xTPjcCS-zoi5zKeL13z0jaG5YXdhITNJREQOtSKBWe4FKU9mQ5-6V3LISiCYygAG5cDme3pbUkZLcD
6. <https://www.sciencedirect.com/science/article/abs/pii/S0021967301008068>
7. Glycans: [https://www.mcponline.org/article/S1535-9476\(20\)35174-4/fulltext](https://www.mcponline.org/article/S1535-9476(20)35174-4/fulltext)

We would like to emphasize that the MSⁿ spectral trees in MSⁿLib provide a unique resource to the metabolomics community. This is the only open, comprehensive dataset containing MSⁿ spectra, considering that mzCloud (provided by HighChem LLC) cannot be downloaded. We are currently designing machine learning models that take MSⁿ trees as input to model the fragment relationships between MS² and higher MSⁿ levels to learn patterns of fragmentation. This is still ongoing research and will still need time to design the best architecture and ML pipeline. However, such research would not be possible without the MSⁿLib library.

The authors still keep an unreadable and uninformative figure, Fig. 3 | Map of the chemical space covered by the MCEBIO MS² library [...]. They now generated more informative Extended Data Fig. 3 | TMAP projections of the chemical space coverage for four mass spectral libraries compared to all chemical structures in those libraries. However, extended data Fig.3 fails to include the MCEBIO library. It would be advantageous to remove Fig and instead give a TMAP projection of the MCEBIO library against the other public libraries.

The reviewer correctly describes the molecular network in Fig. 3 as complex. This figure was added to achieve the following:

1. Provide a quality check on the actual spectral data: Molecular networking will only link entries that contain enough spectral information.
2. See how many spectral entries are “new” or at least different when matched against public libraries: Not only new compound structures are of interest, but also new spectra, acquired with varying collision energies and instruments.
3. Matches to public libraries with perfect MCES = 0 also act as a proof of concept that the workflow selected the correct spectra for compounds that were already present in public libraries.

Fig. 3a gives a general sense that most spectra in this new library contained enough signals to be networked but remained unmatched against all other public spectral libraries. This means they are either completely new structures or at least different enough spectra to add new value to the public domain. Because of the complexity of Fig. 3a, we recommend using the attached Cytoscape network file (Supplementary File 7) for more interactive visualization and a detailed view of the data.

We added a reference to the Cytoscape file to the **Figure 3 caption** to make it more findable: “Overall, most spectra contain enough information to be connected by cosine similarity but remain unmatched against public libraries. The molecular network is shared as a Cytoscape file for interactive exploration (Supplementary File 7).”

The zoomed panel Fig. 3b shows how the Cytoscape file may be used to extract sub-networks that describe a neighborhood or a community of individual nodes or compounds. Nodes with no matches against the public library spectra highlight new spectra, and those with lower MCES scores highlight spectra of new compounds, where public libraries previously captured only similar compounds.

The **Extended Data Fig. 3** shows the chemical space coverage for the complete MSⁿLib. Here, the MCEBIO library is only one of the seven compound libraries included. We only used MCEBIO in Fig. 3 as an illustration of the concept, together with spectral library matching and MCES scoring of the top matches, to reduce the complexity of the visualization. This visualization of spectral quality by molecular networking relies solely on the actual spectral data, whereas the TMAP visualization of **Extended Data Fig. 3** is limited to the structural information without taking the spectral data into account. Therefore, we think that Fig 3 is more valuable than Extended Data Fig. 3 and we prefer to keep both figures in this order. In the last revision, we added the TMAP projections to provide a general sense that all spectral libraries cover various areas of the chemical space. The spectral libraries are rather complementary.

To avoid confusion between MSⁿLib and MCEBIO, we added another TMAP projection as **Extended Data Fig. 3e** showing the coverage for each compound library within MSⁿLib in seven colors.

Extended Data Fig. 3 | TMAP³² projections of the chemical space coverage of MSⁿLib in comparison to other open spectral libraries and the commercial spectral libraries NIST23 and mzCloud. The libraries are divided in a MSⁿLib, containing different compound libraries, b Open libraries, including GNPS, MoNA, and MassBankEU, (access date 08.12.2023) c NIST23 (access date 13.08.2024) and d

mzCloud (access date 04.03.2024). Each node represents an individual structure based on its canonical SMILES representation. Compounds included in the corresponding library are highlighted in magenta and drawn on top of compounds from other libraries in grey. The same structure can be included in multiple libraries. e, demonstrates a more fine-grained MSⁿLib coverage resolved by the individual compound libraries.

The authors asked the reviewer to add a reference that details how MS/MS libraries need to account for impure isomeric compounds (not necessarily only isobaric interferences). Such a reference was recently published as LibGen: Generating High Quality Spectral Libraries of Natural Products for EAD-, UVPD-, and HCD-High Resolution Mass Spectrometers in Analytical Chemistry. 2023 ;95(46):16810-8, but there are surely others.

We thank the reviewer for sharing the reference of LibGen with us. This paper actually caught our attention when it first came out, but we missed citing it in our initial submission. LibGen provides a similar but different workflow for library generation.

We have now added this reference to the main text:

“Consequently, structural isomers, contaminants, and other isobaric interferences may be present during acquisition, annotation, and in the final MSⁿLib. This challenge was already discussed within the LibGen article.(Kong et al. 2023)”

The authors should also include download dates for the different repositories, because they had not used NIST23 before, and it is unclear if other libraries (GNPS, MoNA etc) also had outdated download dates?

We thank the reviewer for the suggestion. The access dates were hidden in the data availability section and the access date for NIST23 was missing, as this dataset is not publicly accessible. For better transparency, we added access dates to the Extended Data Fig. 3. Spectral libraries are growing in size. However, we decided to keep the libraries with the same access date for all figures and analyses as this reflects the state of the data when we first made MSⁿLib publicly available. Redoing all analyses would be a time-consuming effort with little added value as this paper is more focused on the library acquisition method, computational library generation workflow, and the provided resource.

Overall, we already see significant adoption of our MSⁿLib by various groups that develop ML models, many of whom reached out directly after the initial publication of the preprint.

The authors have responded to several of the criticisms that were raised before. They have now added a section on a small use case of drugs metabolites by bacterial communities. The reviewer is not quite clear why this study (MSV000096589) was used, as one would have hoped the authors would use a study with natural compounds (e.g. foods, plants) instead of a drug-based study. The authors confirmed by their stringent criteria (6 matching ions, MS similarity >0.85) that these drugs were indeed added, and stated that existing libraries did not cover these drugs (or did not cover these drugs with those number of matching fragment ions).

While this use case is correctly identifying a few more compounds than before, matching drugs that were missed in classic libraries does not constitute the need for MS³ libraries. The authors do not make a clear case for the use of MS³ libraries, and the reviewer argues that these identifications would have likely been found if those drugs had been represented in classic MS/MS libraries.

Hence, the use case does not provide evidence for the necessity of using a MS³ method as such. Instead, the data provided by the authors only show the need to add more compounds to libraries (which is of course always true).

Less important, the authors argue that Figure 3 is necessary and valid. The presented arguments did not convince the reviewer.

We thank the reviewer for their comments. We used the drug metabolites study, as we included many drugs and drug-like compounds in the first library version. The included compound libraries are based on availability from our collaboration partners. We are certain that, for machine learning models, a broader chemical space, including natural products and synthetic compounds, will increase the suitability of those models. Furthermore, the next steps in leveraging the MSnLib will be in using the MSn data, including the relationships of fragment signals seen in MS² for fragmentation prediction and annotation. This will be a whole new topic and will be done in close collaboration with machine-learning engineers. Based on the transfer to the Brief Communication format, we restructured the paper. Here, we want to focus on the library building workflow and the library's suitability (quality checks, including the drug metabolite study). Additionally, we want to stress the need for collaborations across research topics, broadening the chemical space covered by the public libraries.

brief (maximum 250 characters, including spaces) summary of the main findings (summary in cover letter)

MSⁿLib is an open resource of over 2M multi-stage fragmentation mass spectra covering 30k molecules, including drugs and natural products. The pipeline provides high-throughput data acquisition and processing to automate MSⁿ tree library generation.